# The Fire Modeling Intercomparison Project (FireMIP), phase 1: Experimental and analytical protocols with detailed model descriptions

Sam S. Rabin[1,2], Joe R. Melton[3], Gitta Lasslop[4], Dominique Bachelet[5,6], Matthew Forrest[7], Stijn Hantson[2], Jed O. Kaplan[8], Fang Li[9], Stéphane Mangeon[10], Daniel S. Ward[11], Chao Yue[12], Vivek K. Arora[13], Thomas Hickler[7,14], Silvia Kloster[4], Wolfgang Knorr[15], Lars Nieradzik[16,17], Allan Spessa[18], Gerd A. Folberth[19], Tim Sheehan[6], Apostolos Voulgarakis[10], Douglas I. Kelley[20], I. Colin Prentice[21,22], Stephen Sitch[23], Sandy Harrison[24], and Almut Arneth[2]

[1]Dept. of Ecology & Evolutionary Biology, Princeton University, Princeton, NJ, USA
[2]Karlsruhe Institute of Technology, Institute of Meteorology and Climate Research / Atmospheric Environmental Research, 82467 Garmisch-Partenkirchen, Germany
[3]Climate Research Division, Environment and Climate Change Canada, Victoria, BC, V8W 2Y2, Canada
[4]Land in the Earth System, Max Planck Institute for Meteorology, Bundesstrasse 53, 20146 Hamburg, Germany
[5]Biological and Ecological Engineering, Oregon State University, Corvallis, OR 97331, USA
[6]Conservation Biology Institute, 136 SW Washington Ave., Suite 202, Corvallis, OR 97333, USA
[7]Senckenberg Biodiversity and Climate Research Institute (BiK-F), Senckenberganlage 25, 60325 Frankfurt am Main, Germany
[8]Institute of Earth Surface Dynamics, University of Lausanne, 4414 Géopolis Building, 1015 Lausanne, Switzerland
[9]International Center for Climate and Environmental Sciences, Institute of Atmospheric Physics, Chinese Academy of Sciences, Beijing, China
[10]Department of Physics, Imperial College London, London, United Kingdom
[11]Program in Atmospheric and Oceanic Sciences, Princeton University, Princeton, NJ, USA
[12]Laboratoire des Sciences du Climat et de l'Environnement, LSCE/IPSL, CEA-CNRS-UVSQ, Université Paris-Saclay, F-91198 Gif-sur-Yvette, France.
[13]Canadian Centre for Climate Modelling and Analysis, Environment and Climate Change Canada, Victoria, BC, V8W 2Y2, Canada
[14]Department of Physical Geography, Goethe-University, Altenhöferallee 1, 60438 Frankfurt am Main, Germany
[15]Department of Physical Geography and Ecosystem Science, Lund University, 22362 Lund, Sweden
[16]Centre for Environmental and Climate Research, Lund University, 22362 Lund, Sweden
[17]CSIRO Oceans and Atmosphere, PO Box 3023, Canberra, ACT 2601, Australia
[18]School of Environment, Earth and Ecosystem Sciences, Open University, Milton Keynes, UK
[19]UK Met Office Hadley Centre, Exeter, UK
[20]Centre for Ecology and Hydrology, Maclean building, Crowmarsh Gifford, Wallingford, Oxfordshire, OX10 8BB, UK
[21]School of Biological Sciences, Macquarie University, North Ryde, NSW 2109, Australia
[22]AXA Chair of Biosphere and Climate Impacts, Grand Challenges in Ecosystem and the Environment, Department of Life Sciences and Grantham Institute – Climate Change and the Environment, Imperial College London, Silwood Park Campus, Buckhurst Road, Ascot SL5 7PY, UK
[23]College of Life and Environmental Sciences, University of Exeter, Exeter EX4 4RJ, UK
[24]School of Archaeology, Geography and Environmental Sciences (SAGES), University of Reading, Reading, UK

*Correspondence to:* Sam S. Rabin (sam.rabin@kit.edu)

**Abstract.** The important role of fire in regulating vegetation community composition and contributions to emissions of green-house gases and aerosols make it a critical component of dynamic global vegetation models and Earth system models. Over two decades of development, a wide variety of model structures and mechanisms have been designed and incorporated into global fire models, which have been linked to different vegetation models. However, there has not yet been a systematic examination of how these different strategies contribute to model performance. Here we describe the structure of the first phase of the Fire Model Intercomparison Project (FireMIP), which for the first time seeks to systematically compare a number of models. By combining a standardized set of input data and model experiments with a rigorous comparison of model outputs to each other and to observations, we will improve the understanding of what drives vegetation fire, how it can best be simulated, and what new or improved observational data could allow better constraints on model behavior. In this paper, we introduce the fire models used in the first phase of FireMIP, the simulation protocols applied, and the benchmarking system used to evaluate the models. We have also created supplementary tables that describe, in thorough mathematical detail, the structure of each model.

## 1   Introduction

Several studies have suggested that recent increases in the incidence of wildfire reflect changes in climate (Running, 2006; Westerling et al., 2006). There is considerable concern about how future changes in climate will affect fire patterns (Pechony and Shindell, 2010; Carvalho et al., 2011; Moritz et al., 2012) because of the direct social and economic impacts (Doerr and Santín, 2013; Gauthier et al., 2015), the deleterious effects on human health (Johnston et al., 2012; Marlier et al., 2012), potential changes in ecosystem functioning and ecosystem services (Sitch et al., 2007; Adams, 2013), and impacts through carbon-cycle and atmospheric-chemistry feedbacks on climate (Randerson et al., 2012; Ward et al., 2012; Ciais et al., 2013). Mitigating the most harmful consequences of changing fire regimes – the typical pattern of fire occurrence as characterized by frequency, seasonality, size, intensity, and ecosystem effects, among other factors (Pyne et al., 1996) – could require new strategies for managing ecosystems (Moritz et al., 2014). At the time of the IPCC Fifth Assessment Report, agreement about the direction of regional changes in future fire regimes was considered low – partially as a result of varying projections of future climate (Settele et al., 2014). However, that analysis largely relied on statistical models of fire danger and burned area, forced with a number of different climate projections; the effects of increased atmospheric carbon dioxide, changes in vegetation productivity and structure, and fire-vegetation-climate feedbacks were not considered.

The fact that fire affects so many aspects of the Earth system has provided a motivation for developing process-based representations of fire in Dynamic Global Vegetation Models (DGVMs) and Earth System Models (ESMs). Global fire models have grown in complexity in the two decades since they were first developed (Hantson et al., 2016). The processes represented

– and the forms these processes take – vary widely between global fire models. Although these models generally capture the first-order patterns of burned area and emissions under modern conditions, biases exist in the simulations of seasonality and interannual variability. Evaluating and understanding these differences is a necessary step to quantify the level of confidence inherent in model projections of future fire regimes.

Although it is common practice to compare individual fire models to observations and sometimes previous model versions (e.g., Kloster et al., 2010; Kelley et al., 2013; Yue et al., 2014), no study has directly compared global model performance when driven by the same climate forcing outside the context of model development (i.e., comparing a newly-developed fire module to the one it is designed to replace). One study has performed such a comparison on a regional basis, for Europe (Wu et al., 2015). Less formal comparisons (e.g., Baudena et al., 2015) are difficult to interpret because published simulations

differ in terms of the techniques used to initiate the simulations, the climate inputs used, the time interval considered, and the treatment of land use. Diagnosis of the influence of structural differences between models on simulated fire regimes can only be achieved through a comparison of model performance when forced by identical inputs (e.g., Taylor et al., 2012). The Fire Model Intercomparison Project (FireMIP, http://www.imk-ifu.kit.edu/firemip.php; Hantson et al., 2016) seeks to improve our understanding of fire processes and their representation in global models through a structured analysis of simulations using

identical forcings and the evaluation of these simulations against observations.

  FireMIP will be a multi-stage process. The first stage, described here, will document and investigate the causes of differences between models in simulating fire regimes during the historical era (1901 to 2013). Direct observations of fire occurrence have only been available at a global scale since the 1990s, with the advent of satellite-borne sensors that detect active fires, fire radiative power, and burned area, along with algorithms that automatically process the raw data and output products available

to the general public (Mouillot et al., 2014). Charcoal records do not yet have global coverage, and there are uncertainties even in trend for the twentieth century (Marlon et al., 2016). Literature reviews, sometimes in combination with regional burned area statistics extending back to the 1960s (e.g., Kasischke et al., 2002; Stocks et al., 2003) and/or simulation models, have been used to produce estimates of burned area and associated emissions going back to the beginning of the twentieth century (Mouillot and Field, 2005; Mouillot et al., 2006; Schultz et al., 2008; Mieville et al., 2010). Both remote sensing data and

historical reconstructions can be used to evaluate model performance, but the pre-1990s period – especially before the 1960s – is quite data-poor. This first phase of FireMIP will thus serve to produce an ensemble estimate of global fire activity during that time. Sensitivity experiments will be used to diagnose potential causes of mismatches between simulations and observations. However, fire models can be evaluated only in conjunction with their associated vegetation models: A model that reproduces burned area perfectly but simulates wildly incorrect patterns of aboveground biomass, for example, would be less than ideal.

Likewise, it is possible for biases in a model to cancel each other out, resulting in the right output for the wrong reasons. A number of important vegetation-related variables have observational data available, and FireMIP will assess model simulations of these in addition to fire-related variables so as to holistically evaluate model performance.

  A major goal of FireMIP is to provide well-founded estimates of future changes in fire regimes. In the second phase of FireMIP, we will evaluate how different fire models respond to large changes in climate forcing by running a coordinated

paleoclimate experiment. Past climate states provide the possibility to test the models under environmental conditions against

which they were not calibrated (Harrison et al., 2015), and charcoal records. In this paper, however, we describe the protocol for the first stage of FireMIP: the baseline simulation for the period 1900–2013 and associated sensitivity experiments.

## 2 Experimental protocol

### 2.1 Baseline and sensitivity experiments

The baseline simulation in FireMIP is a fully transient simulation from 1700–2013 (SF1; Table 1). This simulation involves specification of the full set of driving variables and will allow individual model performance to be evaluated against a number of available benchmarking datasets (Sect. 4.1). A series of sensitivity experiments (SF2) will allow the reasons for inter-model agreements and/or discrepancies to be diagnosed by analyzing the impact of each of the main drivers of fire activity separately (Table 1). These experiments use the same input and setup as the SF1 run, but keep key variables constant:

1. "World without fire" (SF2_WWF): Fire is turned off to evaluate the impact of fire on ecosystem processes and biogeography.

2. "Pre-industrial climate" (SF2_CLI): Climate forcings are fixed to repeated 1901–1920 levels to analyze the impact of historical climate changes on photosynthesis and consequent impacts on fire and other ecosystem processes.

3. "Pre-industrial $CO_2$" (SF2_CO2): Atmospheric $CO_2$ concentration is fixed to pre-industrial levels (277.33 ppm) to analyze the impact of historical $CO_2$ increases on photosynthesis and consequent impacts on fire and other ecosystem processes.

4. "Fixed lightning" (SF2_FLI): Historically-varying lightning data are replaced with repeated cycles of lightning from 1901–1920 to explore the impact of changes in this potentially important source of ignitions.

5. "Fixed population density" (SF2_FPO): Human population density is fixed at its value from 1700, humans being another important source of ignitions whose distribution and number has changed over the last three centuries.

6. "Fixed land use" (SF2_FLA): Distributions of cropland and pasture are fixed at 1700 values to assess the impacts of historical land use changes and inter-model differences in implementation.

Limitations related to model structure and other constraints mean that not all participating models will be able to perform every SF2 experiment.

### 2.2 Input datasets

The FireMIP baseline experiment is driven by a set of standardized inputs, which include climate, population, land use and lightning. The climate forcing is based on a merged product of Climate Research Unit (CRU) observed monthly 0.5° climatology (1901–2013; Harris et al., 2014) and the high temporal resolution NCEP reanalysis. The merged CRU-NCEP v5 product

has a spatial resolution of 0.5° and a 6-hourly temporal resolution (Wei et al., 2014). Global atmospheric $CO_2$ concentration was derived from ice core and NOAA monitoring station data (Le Quéré et al., 2014) and is provided at annual resolution over the period 1750–2013.

Many of the participating models were developed using different climate forcing data. Figure 1 illustrates how serious an impact this can be, using the JSBACH-SPITFIRE fire model (Lasslop et al., 2014). This model configuration was originally parameterized using the CRU-NCEP forcing data. When the CRU-NCEP wind forcing is substituted with that from the WATCH data (Weedon et al., 2011), modeled burned area decreases by ca. 27% with important spatial changes in regional patterns. Because the use of different input data – in this case wind speed – can produce such major differences in outputs, participating groups were allowed to reparameterize their fire models to adjust for the idiosyncrasies of the FireMIP-standardized input data.

Annual data from 1700–2013 at 0.5° resolution on the fractional distribution of cropland, pasture, and wood harvest – as well as transitions among land use types – were taken from the data set developed by Hurtt et al. (2011). This data set is based on gridded maps of cropland and pasture from version 3.1 of the History Database of the Global Environment (HYDE; Klein Goldewijk et al., 2010), which are generated based on country-level FAO statistics of agricultural area in combination with algorithms to estimate population, land use, and settlement patterns into the past. HYDE also provides gridded maps of historical population density, which participating FireMIP groups used if needed.

A global, time-varying data set of monthly cloud-to-ground lightning was developed for this study at 0.5° and monthly resolution (J. Kaplan, personal communication), comprising global lightning strike rate (strikes $km^{-2}$ $day^{-1}$), for the period 1871–2010. This dataset incorporates interannual variability in lightning activity using the method described by Pfeiffer et al. (2013) by scaling a mean monthly climatology of lightning activity (covering 2005–2014; Virts et al., 2013) using convective available potential energy (CAPE) anomalies (Compo et al., 2011).

The participating models (Table 2) have different spatial and temporal resolutions; groups were thus allowed to interpolate inputs from their original resolution to that appropriate for their model. This was done so as to preserve totals as close as possible to the canonical data. Some models required additional input datasets – for example, nitrogen deposition rates or soil properties. These were not standardized.

## 2.3 Model runs

The models were spun up to a pre-industrial equilibrium state. For these spin-up runs, population density and land use were set to their values in 1700 CE, and atmospheric $CO_2$ concentration was set to its year 1750 CE value of 277.33 ppm. Climate and lightning forcings from 1901–1920 were used, being recycled until carbon values in the slowest soil carbon pool varied by less than 1% between consecutive 50-year periods for every grid cell (Fig. 2). Note that for various reasons some modeling groups may not be able to use 1700 CE as the beginning of their run, with CLM-Li preferring 1850 and CTEM preferring 1861.

The historic simulations were run from 1700 through 2013. Population and land use were changed annually from the beginning of this simulation, and $CO_2$ values were changed annually from 1751 onwards. However, because the CRU-NCEP and lightning forcing data were not available for 1700–1900, the 1901–1920 forcings were recycled for the first 200 years of the simulation; this allowed natural climate variability to be captured while incorporating only minimal human influence. From

1901 to 2010, time-varying values of all variables were used. Finally, the lightning dataset did not include 2011–2013, so the 2010 values were used for the last three years of the experiment. A visualization of the time periods covered by each input in the spinup and historical model runs can be found in Figure 2.

Although agriculture (cropland and pasture) were specified inputs, each model calculated natural vegetation on other grid
cells according to its standard set-up and no attempt was made to standardize this. The biogeography of natural vegetation, represented by plant functional types (major global vegetation classes; PFTs), was either prescribed by modeling groups or simulated dynamically (Table 2).

## 2.4   Output variables

A basic set of gridded outputs (Table 3) covering the period 1950–2013 is required for model comparison and evaluation. An
additional set of output variables (Table A1) is provided for diagnostic purposes. All outputs are to be provided in NetCDF format at the native spatial resolution of the model, and at either monthly or annual temporal resolution (Tables 3, A1). In addition to the gridded outputs, global total fire emissions per year from the period 1700–2013 are to be provided in ASCII format.

## 3   Participating models

Eleven models are running the phase 1 FireMIP simulations (Table 2). All simulate fire in "natural" ecosystems, which are composed of a variety of PFTs representing major vegetation classes around the world. Some models also simulate cropland, pasture, deforestation, and peat fire (Table S3). Figures 3–5 use the metaphor of a flowchart to illustrate the differences among the fire models in terms of structural organization and process inclusion. Whereas LPJ-GUESS-GlobFIRM and LPJ-GUESS-SIMFIRE-BLAZE use relatively simple empirical models to estimate gridcell burned area directly, the other models use a
process-based structure to separately simulate fire occurrence (Fig. 3) and burned area per fire (Fig. 4). Even within the process-based models, however, a wide range of complexity is evident. For example, the calculation of burned area per fire (Fig. 4) can be as simple as the PFT-specific constants used in JULES-INFERNO, or can be so complex as to consider factors such as human population density and economic status, fuel moisture and loading, and wind speed. Translating from burned area to effects on the ecosystem shows a similar variation in model strategy, although models tend to fall into two groups (Fig.
5). Some models define constant combustion and mortality factors to calculate the fraction of vegetation burned or killed in a fire, whereas the rest – JSBACH-SPITFIRE, LPJ-GUESS-SIMFIRE-BLAZE, LPJ-GUESS-SPITFIRE, LPJ-LMfire, MC-Fire, and ORCHIDEE-SPITFIRE – vary fractional mortality and combustion based on estimated fire intensity, PFT-specific plant architecture and fire resistance, and other factors.

The models also differ in the order in which fire-affected live biomass is combusted (transferred to the atmosphere) and
killed (transferred to soil and/or litter pools; Fig. 5, Tables S12–S13). CLM-Li, LM3-FINAL, LPJ-GUESS-SPITFIRE, and ORCHIDEE-SPITFIRE combust live biomass first, then apply fire mortality to the remaining non-combusted biomass. JSBACH-SPITFIRE, LPJ-GUESS-GlobFIRM, LPJ-GUESS-SIMFIRE-BLAZE, LPJ-LMfire, and MC-Fire, on the other hand, first "kill"

biomass, then apply combustion to that killed fraction; the remaining non-combusted fraction of "killed" biomass is transferred to litter or soil pools (i.e., experiences mortality as defined here). CTEM calculates both combustion and mortality as fractions of pre-burn biomass.

A more detailed and mathematical description of the fire models can be found in Tables S1–S28. In these, to the extent possible, we have included all the equations and parameters used by each model to calculate burned area and fire effects. Based on model descriptions available in the literature, combined with unpublished descriptions, model code, and extensive conversations with developers, these tables represent the most complete description yet of the inner workings of several fire models. Units have been standardized, variable names have been harmonized, and analogous processes have been grouped together. We have also included PFT-specific parameters and equations in Tables S17–S28; these were prescribed by the modeling groups during the development of their respective fire models either due to limitations of their vegetation models or intentionally based on development plans and priorities. Together with Figures 3–5, the tables enable the straightforward comparison of models whose published descriptions often do not adhere to the same conventions, and will be important tools in interpreting inter-model variation in the results of the experiments described in this paper. They will also prove useful for other researchers interested in how global fire models work and how they differ from each other. It should be noted, however, that most of these models are under continuous development; it should not be assumed that the descriptions given here apply to anything except the model versions used for this phase of FireMIP.

In this section, we briefly describe each participating model, including details of how the model versions used for FireMIP differ from any published versions.

### 3.1 CLM fire module

The fire model described by Li et al. (2012, 2013, 2014), with adjusted fuel moisture parameters (Li and Lawrence, 2017), was used in the NCAR CLM4.5-BGC land model (Oleson et al., 2013) to provide outputs for FireMIP. This model includes empirical and statistical schemes for modeling burned area of and emissions from crop fires, peat fires, and deforestation and degradation fires in tropical closed forests. A process-based fire model of intermediate complexity simulates non-peat fires outside croplands and tropical closed forests. CLM4.5-BGC does not output fire counts and fire size because the two variables are not used in the schemes for crop fires, peat fires, and deforestation and degradation fires in tropical closed forests. Note that this fire model does not simulate fireline intensity. In addition, CLM4.5-BGC does not distinguish above-ground and below-ground litter (Koven et al., 2013). For simplicity, this model may be referred to as CLM-Li, or CLM-Li* when only referring to the model for non-peat fires outside croplands and tropical closed forests.

### 3.2 CTEM fire module

The Canadian Terrestrial Ecosystem Model (CTEM v. 2.0; Melton and Arora, 2016) represents disturbance as both natural and human-influenced fires. The original fire parameterization is described in Arora and Boer (2005), with Melton and Arora (2016) describing recent changes and its implementation in CTEM v. 2.0. The only changes between the version of the model

used here and that described by Melton and Arora (2016) are for the vegetation biomass thresholds for fire initiation ($S_L$; Table S4) and the PFT-specific fractional combustion of leaves ($\widehat{FC_{l,leaf}}$), stems ($\widehat{FC_{l,stem}}$), and litter ($\widehat{FC_{d,litter}}$; see Table S16).

## 3.3  JULES-INFERNO

The INteractive Fire and Emission algoRithm for Natural envirOnments (INFERNO; Mangeon et al., 2016) was developed for
the UK Met Office's Unified Model (UM) and has been integrated within the Joint UK Land Environment Simulator (JULES; Best et al., 2011; Clark et al., 2011). JULES-INFERNO focuses on offering a simple, stable parameterization to diagnose fire occurrence, burnt area, and biomass burning emissions in the context of an Earth system model. It builds upon the fire parameterization proposed by Pechony and Shindell (2009). It is an empirical scheme that uses vapor pressure deficit (Goff and Gratch, 1946), precipitation, and soil moisture to diagnose burnt area and subsequent emissions. Within JULES-INFERNO,
humans only explicitly impact biomass burning through the number of fires. The algorithm foregoes physical calculations for the rate of spread, instead assigning a vegetation-dependent average burned area: 0.6, 1.4, and 1.2 $\mathrm{km}^2$ for fires in trees, grasses, and shrubs, respectively. Because of this specificity, no outputs for fire counts and fireline intensity are provided. Furthermore, fire-induced tree mortality and vegetation carbon removal have not been included. The FireMIP simulations were run on a relatively coarse N96 grid (192 cells longitude by 145 cells latitude).

## 3.4  JSBACH-SPITFIRE

The SPITFIRE model (Thonicke et al., 2010) was implemented in the JSBACH land surface component of the MPI Earth System Model (MPI-ESM; Giorgetta et al., 2013) to account for the effect of fire on vegetation, the carbon cycle, and the emissions of trace gases and aerosols into the atmosphere. The resulting JSBACH-SPITFIRE model (Lasslop et al., 2014) runs on a daily time step and can be applied in a coupled MPI-ESM model setup as well as an offline model forced with
meteorological input data. Differences between JSBACH-SPITFIRE and the original SPITFIRE model described by Thonicke et al. (2010) include a modification of the effect of wind speed on fire spread rate, changes to parameters related to human ignitions and fuel drying, and a dependence of fire duration on population density (Lasslop et al., 2014). There have been several as-yet-unpublished changes to JSBACH. The conversion factor from biomass to carbon was changed from 0.45 to 0.5 to ensure consistency with emission factors. The definition of the green pool was revised to include only 1-hour fuel, while
previously it also included sapwood. Finally, combustion completeness has been changed to match to that used by ORCHIDEE-SPITFIRE (Yue et al., 2014), which are based on a recent collection of field measurements (van Leeuwen et al., 2014).

## 3.5  LM3-FINAL

The Fire Including Natural & Agricultural Lands model (FINAL; Rabin, 2016; Rabin et al., in prep.) simulates global fires within the Geophysical Fluid Dynamics Laboratory Land Model version 3 (LM3; Shevliakova et al., 2009; Milly et al., 2014;
Sulman et al., 2014). FINAL follows the structure of Li et al. (2012, 2013) closely for prediction of wildland fires with lightning and human ignition, but does not have special modules for deforestation and peatland fires. Previous work (Magi

et al., 2012; Rabin et al., 2015) estimated the amount of burned area from cropland, pasture, and non-agricultural fires based on total observed burned area from the Global Fire Emissions Database version 3 including small fires (GFED3s; Randerson et al., 2012). The non-agricultural burned area estimates from Rabin et al. (2015) serve as the basis for parameter estimation in FINAL, which is accomplished using an implementation of the Levenberg-Marquardt method (Rabin, 2016; Rabin et al., in prep.). Cropland and pasture fires are computed on a monthly basis, based on regional climatologies of burned fraction derived from a statistical analysis of observed burning and land cover distributions Rabin et al. (2015). The version of FINAL used here enhances rate of spread in crown fires relative to surface fires; these are distinguished using predictions of fireline intensity and vegetation height. In addition, this version of FINAL uses fire termination conditions to determine fire duration; whereas fire duration was previously fixed at one day, that is now the minimum. Lastly, parameters are optimized separately for boreal climate zones and non-boreal climate zones. Similarly to CLM-Li*, here LM3-FINAL* will refer to the fire model on non-agricultural land.

### 3.6 LPJ-LMfire

The LPJ-LMfire model (Pfeiffer et al., 2013) is based on the SPITFIRE model (Thonicke et al., 2010) with a number of modifications to improve the simulation of fire starts, fire behavior, and fire impacts. LPJ-LMfire was specifically designed for the simulation of fire in preindustrial time, and specifies the ways in which humans use fire based on their subsistence livelihood, breaking populations into three categories: hunter-gatherers, pastoralists, and farmers. The model accounts for feedbacks between human agency and biogeography, in particular in the way that hunter-gatherers can increase the carrying capacity of their environment through the managed application of fire, i.e., niche construction. LPJ-LMfire also simulates passive fire suppression due to landscape fragmentation, assuming that agricultural land is not subject to wildfire. LPJ-LMfire was used to simulate the impact of humans on continental-scale landscapes during the Last Glacial Maximum (Kaplan et al., 2016) and in late preindustrial time (Hopcroft et al., 2017). In contrast to LPJ-GUESS-SPITFIRE, LPJ-LMfire runs in "population mode," where vegetation is represented by "average individuals" as opposed to cohorts. This necessitated some enhancements to LPJ beyond the fire model itself, including a simplified representation of vegetation structure achieved by disaggregating average individuals into height classes. For the FireMIP experiments described in this paper, we used LPJ-LMfire v1.0 as described in Pfeiffer et al. (2013) without modifications. However, to provide a bracketing scenario of anthropogenic ignitions, we performed contrasting simulations where farmers and pastoralists either ignited fire according to our standard preindustrial formulation, or not ignite any fire at all.

### 3.7 LPJ-GUESS-GlobFIRM

The Lund-Potsdam-Jena General Ecosystem Simulator (LPJ-GUESS) dynamic global vegetation model includes the Glob-FIRM fire model (Thonicke et al., 2001) to estimate global fire disturbance. GlobFIRM simulates fire once per year if enough fuel is available, with annual fire probability based on the daily water status of the upper soil layer over the previous year. Fuel consumption and vegetation mortality then depend on fire probability and a PFT-specific fire resistance parameter. (As LPJ-GUESS-GlobFIRM estimates burned area directly, it does not generate outputs of fire count or size.) While LPJ-GUESS shares

many core ecophysiological features with the other models in the LPJ family (Sitch et al., 2003), its distinguishing feature is that it also includes detailed representations of stand-level vegetation dynamics (Smith et al., 2001). In LPJ-GUESS, these are simulated as the emergent outcome of growth and competition for light, space, and soil resources among annual cohorts of woody plants and an herbaceous understory (Smith et al., 2001). These processes are simulated stochastically by using multiple

"patches," each representing random samples of each simulated locality or grid cell and which correspond to different histories of disturbance and stand development (succession). Recently, the nitrogen cycle and N limitations on primary production were included in LPJ-GUESS (Smith et al., 2014), as well as land management for pastures and croplands (Lindeskog et al., 2013).

## 3.8  LPJ-GUESS-SIMFIRE-BLAZE

The new BLAZe-induced land-atmosphere flux Estimator (BLAZE; Nieradzik et al., in prep.) was recently implemented into
10 the latest version of LPJ-GUESS (Lindeskog et al., 2013; Smith et al., 2014). Burned area is generated once per year by the empirical fire model SIMFIRE (Knorr et al., 2014, 2016) based on fire weather, fuel continuity, and human population density. This annual burned area is distributed to each month of the year based on mean observed seasonality (climatology) of burned area from GFED3 (Giglio et al., 2010). Fuel consumption and tree mortality are then estimated using the BLAZE module, which computes fire-line intensities from existing fuel load and fire weather parameters which are translated into height-
15 dependent survival probabilities as described in the Population-Order-Physiology (POP) tree demography model (Haverd et al., 2014). Mortality functions for different biomes are derived from the literature (Hickler et al., 2004; van Nieuwstadt and Sheil, 2005; Kobziar et al., 2006; Bond, 2008; Dalziel and Perera, 2009). The fluxes between live and litter pools and the atmosphere are then computed accordingly.

## 3.9  LPJ-GUESS-SPITFIRE

The SPITFIRE model (Thonicke et al., 2010) was originally added to the LPJ-GUESS vegetation model (Ahlström et al., 2012) by Lehsten et al. (2009, 2015). This implementation generally followed the original SPITFIRE formulation, but initial applications employed prescribed fire regimes and did not use the full set of burned area calculations in SPITFIRE. This initial version also included modifications to account for the detailed representation of stand-level vegetation dynamics in LPJ-GUESS. For example, because many patches are smaller than many individual fires, each patch burns stochastically at
each time step, with the probability of a patch burning set equal to the gridcell burned fraction in that time step. The version of LPJ-GUESS-SPITFIRE used here extends the version of Lehsten et al. (2009, 2015) by incorporating the complete burned area calculation from SPITFIRE (Thonicke et al., 2010), including lightning ignitions, burnt area, fire intensity, residence time, and trace gas emissions. However, human ignitions have been recalibrated to match global burned area data, and the effect of wind speed on rate of spread has been modified (Lasslop et al., 2014). The raingreen phenology follows Lehsten et al. (2009, 2015)
and the PFT parameterization follows Forrest et al. (2015), but some important parameters for post-fire mortality and biomass of tropical trees have been updated since those publications. These are: tree allometry (Feldpausch et al., 2011; Dantas and Pausas, 2013), bark thickness (Mike Lawes, unpublished data), fuel bulk density (from Hoffmann et al., 2011), and maximum crown area (increased to 300 $m^2$ based on Seiler et al., 2014, but taking a more conservative value appropriate for a global

parameterization). For details see Table S22. Furthermore, a simple land use scheme was implemented for compliance with the FireMIP protocol. A time-evolving fraction of patches was designated as pasture or cropland based on the HYDE land use data set (Klein Goldewijk et al., 2010). When natural patches were converted to cropland or pastures, 90% of the aboveground carbon was immediately respired to the atmosphere and 10% was added to a woody products carbon with a 25-year residence time (following Lindeskog et al., 2013). In cropland and pasture patches, tree establishment is forbidden, so only grass PFTs are present. Lightning ignitions occur in both cropland and pasture, but human ignitions were forbidden in croplands. One further change to the model compared to previous versions is that fuel moisture was taken as the average of the standard SPITFIRE fuel moisture (calculated per fuel class based on a fire danger index) and soil moisture. This was done to take into account the vertical moisture gradient through the fuel bed from the topmost fuel (whose moisture will equilibrate with the air moisture) and the bottommost fuel (which will be in contact with the soil and therefore will tend to equilibrate with soil moisture). This improved the timing and magnitude of simulated burnt area in development simulations.

## 3.10 MC-Fire

The MC-Fire module (Conklin et al., 2015; Lenihan and Bachelet, 2015) simulates fire occurrence, area burned, and fire impacts including mortality, consumption of aboveground biomass, and nitrogen volatilization. Mortality and consumption of overstory biomass are simulated as a function of fire behavior and the canopy vertical structure. Fire occurrence is simulated as a discrete event, with an ignition source assumed to always be present and generating at most one fire per year in a grid cell. Fire return interval varies between minimum and maximum values for each vegetation type, based on fuel loading and moisture. The version of MC-Fire run here is identical to the version described by Conklin et al. (2015) and Lenihan and Bachelet (2015).

## 3.11 ORCHIDEE-SPITFIRE

The ORCHIDEE-SPITFIRE model was developed by incorporating the SPITFIRE model (Thonicke et al., 2010) into the land surface model ORCHIDEE. All equations as described in Thonicke et al. (2010) were implemented, except for changes to lightning ignitions and combustion completeness, as well as the addition of a fuel-dependent ignition efficiency term (as described in Yue et al., 2014, 2015). Combustion completeness values were updated compared to those in Yue et al. (2014, 2015), based on data published in van Leeuwen et al. (2014). Regional scaling factors for burned area were also introduced, to adjust simulated regional burned area for 1997–2009 to agree with that reported in version 3 of the Global Fire Emissions Database (GFED3; Giglio et al., 2010). The regions used were the 14 GFED regions (van der Werf et al., 2006). Finally, the standard FireMIP lightning dataset was adjusted to account for the fact that the original model (Yue et al., 2014, 2015) was calibrated using the LIS/OTD lightning flash rate climatology (Cecil et al., 2014, http://gcmd.nasa.gov/records/GCMD_lohrmc.html). Specifically, the cloud-to-ground numbers provided were scaled to total (i.e., cloud-to-ground plus within-cloud) flashes, so that the mean annual global lightning flash rate during 1997–2009 was the same as that given in the LIS/OTD data.

## 4 Model evaluation

### 4.1 Benchmarking protocol

The mean and variance of global agreement between model and observations provide basic measures of model performance. Model outputs will be compared to observations using the metrics devised by Kelley et al. (2013) to quantify model perfor-
5 mance for individual processes. This system uses normalized mean error (NME) and normalized mean squared error (NMSE) to evaluate geographic patterns of total values, annual averages, and interannual variability. Spatial performance of variables measuring relative abundance (i.e., cases where the sum of items in each cell must be equal to one, as in the case of vegetation cover) are evaluated using the Manhattan Metric (MM) or squared chord distance (SCD). Kelley et al. (2013) also developed metrics to assess temporal performance – for example, comparing the timing and length of the simulated fire season,
and the magnitude of differentiation between seasons – with observations. These standardized statistics allow straightforward comparison of model performance with regard to variables that may have differences in units of many orders of magnitude.

Kelley et al. (2013) also introduced the idea of creating a kind of statistical control for putting these metric scores into context. The "mean model" consists of a dataset of the same size as the observations, where every element is replaced with the observational mean. Similarly, the "random model" is produced by bootstrap resampling of the observations. These datasets
allow the performance of the actual models to be compared against an external standard in addition to each other for individual processes of interest. If a model does not perform significantly better than one using the mean or random data, its usefulness may be limited. Additionally, as the metrics used represent normalized "distance" between models and observations, a comparison of scores shows how much closer to reality one model is than another. For example, a model's score of 0.5 is exactly 33% closer to the observations than another of 0.75 (0.5/0.75 = 33%). Conversely, the second model would need to improve
by 33% in order to provide as good a match to observations as the first.

This benchmarking system can be used to evaluate model performance with regard to aspects of land and vegetation other than fire. In addition to burned area and fire emissions, we will use observational datasets of vegetation properties and hydrology to evaluate how well the models simulate the land-vegetation system as a whole. This is especially important because burning affects a wide range of Earth system processes, often in a non-linear manner.

Following the procedure described by Kelley et al. (2013) will help quantify the spatial and temporal biases in mean and variability of a range of variables important to the Earth system. Diagnosing the ultimate causes of those biases is problematic due to the myriad interactions between fire, vegetation, and the atmosphere. Only targeted experiments will allow sufficient process isolation to provide controlled tests of the importance of different mechanisms. The SF2 experiments, in which certain processes are fixed or disabled, represent a first step in this direction. The analysis described for this first phase of FireMIP will
likely highlight other inter-model differences that have significant impacts on performance, with the purpose of serving as a jumping-off point for further experimentation and development.

The complete set of observational datasets to be used in this phase of FireMIP can be found in Table 4, and a description of the criteria for choosing datasets is given in Section 4.3 below.

## 4.2 Comparison to empirical relationships

Benchmarking will establish the degree to which a model is able to reproduce key temporal and spatial patterns in fire regimes and drivers of fire regimes, including vegetation and hydrology. However, it is important to establish that the model reproduces these patterns for the right reasons rather than because it is highly tuned. Analyses involving process evaluation focus on assessing the realism of model behavior rather than simply model response, a necessary step in establishing confidence in the ability of a model to perform well under substantially different conditions from present. The basis of such analyses is the identification of relationships between key processes and potential drivers, based on analyses of observations using tools such as generalized linear models (GLMs) to isolate meaningful relationships (e.g. Daniau et al., 2012; Bistinas et al., 2014). Model outputs can then be interrogated to determine whether the model reproduces these relationships (e.g., Lasslop et al., 2014; Li et al., 2014). We plan to apply GLMs to both observational datasets and to the corresponding model forcing variables and model outputs to identify relationships between fire activity and potential climatic, vegetative, and socio-economic drivers. This will allow us to analyze the sensitivity of the simulated fire activity to various controls, as well as to evaluate how well the models recreate emergent relationships seen in observational data.

## 4.3 Observational data

The observational database assembled for FireMIP consists of a collection of datasets selected to allow systematic evaluation of a range of model processes. The system is an updated and extended version of that presented by Kelley et al. (2013). As in Kelley et al. (2013), the site-based and remotely-sensed observational data sets were chosen to fulfill a number of criteria. They are all global in coverage or provide an adequate sample of different vegetation types on each continent. The datasets are also all independent, in that they do not require the calculation of vegetation properties from the same driving variables as the fire-enabled DGVMs. This excludes, e.g., net primary productivity or evapotranspiration products that are based on the interpretation of remotely-sensed data using a vegetation model. For variables that display significant seasonal or interannual variability, the data must be available for multiple years and seasonal cycles. And finally, the data must be publicly accessible, so that other modeling groups can use the benchmarks subsequently.

The selected datasets provide information for vegetation properties, fire properties, hydrology, and fire emissions (Table 4). All remotely sensed data were re-gridded to a $0.5°$ grid and masked to a land mask common to all the models. There are multiple data sets available for some variables; we retained all of these products in order to be able to take account of observational uncertainties in the benchmarking procedure. It should be noted that many of the individual data sets do not provide measures of uncertainty.

The analytical protocol we have described is appropriately rigorous and transparent. However, the effectiveness of any model evaluation is dependent on the quality of its observational data, and FireMIP is no different. There are no data available at scales relevant to global models for an number of important fire-related variables – for example, ignition frequency or fraction of trees killed by fire. The variables that do have global data from remote sensing often suffer from substantial uncertainty, as discussed for burned area by Hantson et al. (2016).

# 5 Discussion and Conclusions

The goal of FireMIP is to compare the performance of a number of different global fire models in a systematic and uniform manner, evaluating model performance against standard benchmarks. Each model has been developed for different purposes, and thus we cannot expect that they will be equally good at simulating every aspect of the fire regime. Thus, our goal is not to identify a single best model, but rather to assess the strengths and weaknesses of individual models, and to identify how individual models could be improved.

The FireMIP protocol uses standardized inputs for climate, lightning, land use, and population density. These inputs represent major drivers of fire regimes, and standardization should therefore minimize a major cause of differences between model simulations and help to isolate the impact of structural differences between the models on the simulation of fire regimes. However, there are secondary sources of inter-model differences that are more difficult to standardize and are not dealt with in this protocol. For example, each of the models prescribes or simulates natural vegetation outside of agricultural and/or urban areas. Differences in the prescribed or simulated natural vegetation at a regional scale will lead to differences in the simulated fire regimes. However, prescribing vegetation distributions in coupled fire-vegetation models means neglecting the critical two-way interaction between vegetation type and fire regime, and real-world interactions between climate and the coupled fire-vegetation system conflict with the idea of prescribing vegetation in Earth system models. Outputting information on leaf area and fractional cover of different PFTs (Table 3) will, at least, make it possible to examine whether differences in the simulated regional fire regimes reflect significant differences in vegetation. Similarly, the protocol has not standardized soil inputs – which will affect the water-balance calculations and hence control vegetation distribution – because this would likely require major re-calibrating of the models. However, differences in the soil inputs used by individual models could lead to differences in fire regimes at a regional scale. We anticipate that this is a second-order effect, and will rely on process-based diagnoses to identify the degree to which it explains inter-model differences. Finally, the exact implementation of land use and land cover change can cause important differences in model outputs, even given the same land use driver dataset (Brovkin et al., 2013).

The participating models vary in spatial resolution: Most are run on a 0.5° grid but some are run at coarser resolution (Table 2) and provide outputs at the native resolution of the model. Model parameterizations are specific to model resolution, and thus differences caused by differences in resolution are an inherent part of the structural uncertainty. However, resolution has an impact on the benchmarking metrics, with goodness-of-fit being inflated as resolution becomes coarser. Thus, the interpretation of the benchmarking metrics will need to take this into account by calculating appropriate null models for the different resolutions.

Model benchmarking will examine several different aspects of the fire regime, but will also consider how well each model captures vegetation properties and hydrology (Table 4). There are multiple data sets available for some of these properties, including, for example, burned area. Padilla et al. (2015) have shown that currently-available burned area products differ considerably both in terms of global total and at a regional scale. Differences between different data sets effectively define the

current range of uncertainty in observations, and this level of uncertainty needs to be taken into account when evaluating model performance.

Eleven modeling groups are performing the baseline FireMIP simulations, but there are several other fire models in use. We hope that publishing this experimental and benchmarking protocol will encourage other fire modeling groups to participate in FireMIP.

We provide a standardized modeling and benchmarking protocol for a wide variety of global fire-enabled ecosystem models. The wide variety of approaches taken by the participating models lead us to expect notable inter-model variation in results. Some models, for example, estimate energy release for calculations of fire behavior and effects, while others use simplifications – an important structural difference. Process treatment (and, indeed, inclusion) should also cause variation in results; human ignitions and suppression, for example, are treated very differently by the different models, with some ignoring them entirely. By systematically comparing models developed with such a wide array of approaches, this effort will advance our understanding of fire dynamics and its effects on ecosystem and Earth system functioning. The analyses will reveal important model shortcomings, which are crucial for assessing model uncertainties in future projections, and should, in the longer term, contribute to the development of better and more reliable fire models and projections.

**Data availability**

Once all runs are completed, model outputs will be made available to the public at https://bwfilestorage.lsdf.kit.edu/public/projects/imk-ifu/FireMIP. The FireMIP website (http://www.imk-ifu.kit.edu/firemip.php) will also be kept up-to-date with the latest data access details in addition to project updates and summary information.

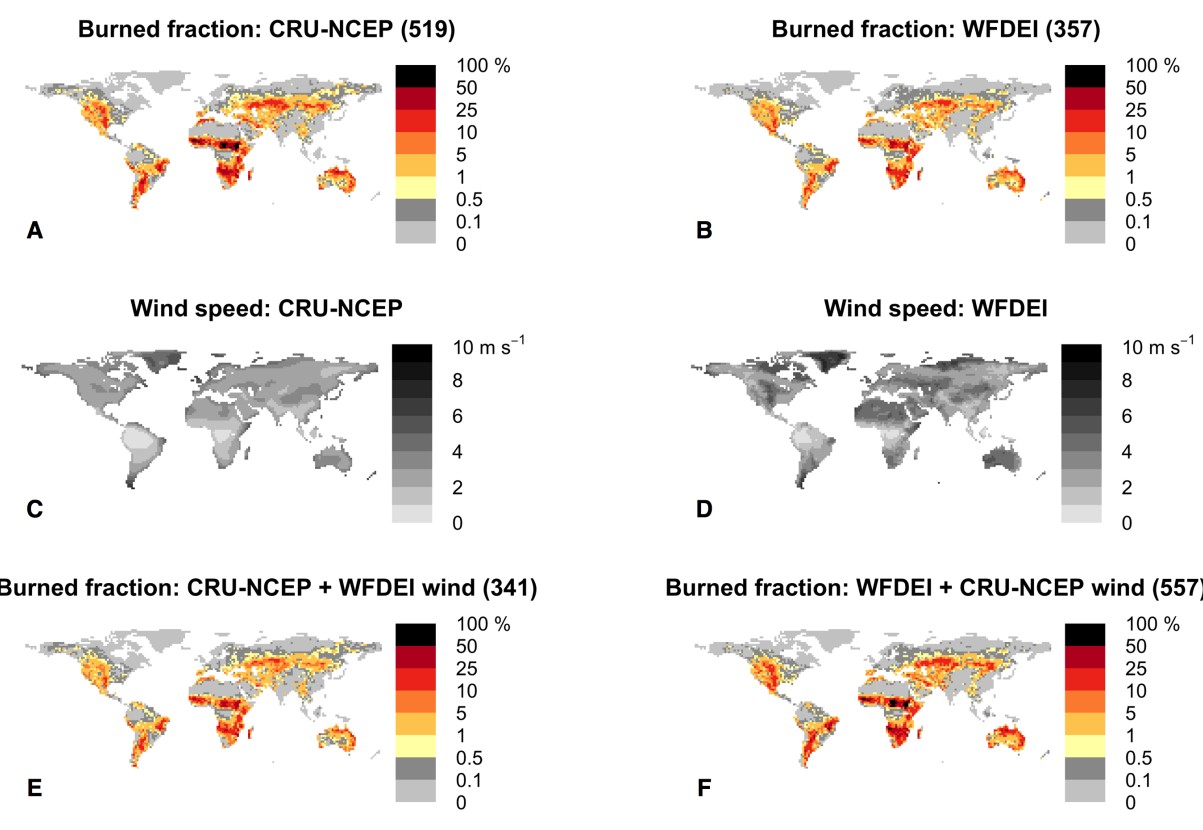

**Figure 1.** Comparing the effect of different wind forcing data on burned area simulated by JSBACH-SPITFIRE (Lasslop et al., 2014) over the years 1997–2005. (**a–b**) Annual burned fraction (%) modeled by JSBACH-SPITFIRE using (**a**) the CRU-NCEP forcing data (Wei et al., 2014) and (**b**) the WATCH (WFDEI) forcing data (Weedon et al., 2011). (**c–d**) Mean wind speed over the simulated period from (**c**) the CRU-NCEP and (**d**) WFDEI datasets. (**e–f**) Annual burned fraction (%) modeled by JSBACH-SPITFIRE with switched wind forcing: (**e**) CRU-NCEP except with WFDEI wind, (**f**) WFDEI except with CRU-NCEP wind. Numbers in sub-figure titles give mean annual global burned area (Mha) for each run.

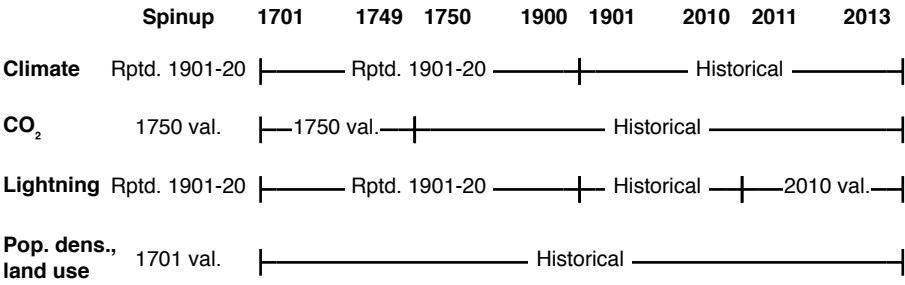

**Figure 2.** Timelines describing how the different input datasets were used in the spinup and historical model runs. X-axis not to scale. "Historical": Time series of observation-based data. "Rptd. 1901–20": Repeated time series of values from 1901–1920. "YEAR val.": Variable held constant at value for year YEAR.

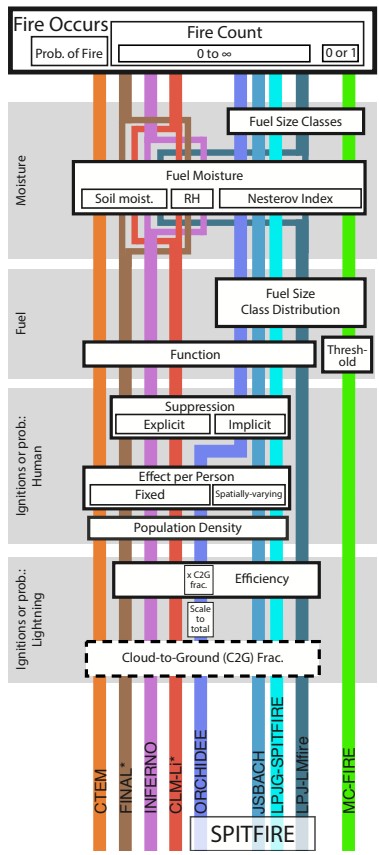

**Figure 3.** Modeled processes leading to fire starts for the participating models. Beginning at the bottom, models explicitly simulate processes that their colored line passes through, with the end result being the calculation of fire count (which in most models can be any nonnegative number, but in MC-FIRE can only be zero or one) or probability of fire. (LPJ-GUESS-SIMFIRE-BLAZE and LPJ-GUESS-GlobFIRM are not included here because they do not calculate fire count or probability.) Fire occurrence depends on three factors: Ignitions, fuel availability, and fuel moisture. Lightning ignition count or probability are functions of the flash rate multiplied in some models by the "cloud-to-ground fraction" (which the input data for FireMIP already includes and is thus not calculated here; dashed box) and/or by an "Efficiency" term describing what fraction of cloud-to-ground strikes actually serve as potential ignitions. (ORCHIDEE-SPITFIRE scales cloud-to-ground flash rate to total flash rate, then multiplies by a coefficient representing both cloud-to-ground fraction and ignition efficiency.) Human ignition count or probability are influenced by an "effect per person" parameter, which can either be "fixed" globally or "spatially-varying." Population density can also contribute to "suppression." Suppression as a function of population density can be either "explicit" (i.e., calculated by a specific function) or "implicit" (i.e., included in the initial calculation of ignitions/probability). Fuel load affects fire occurrence either as a simple "threshold" or by the use of some more complex "function" such as a logistic curve. Some models use several "fuel size classes," which can be important for both fuel loading and moisture terms.

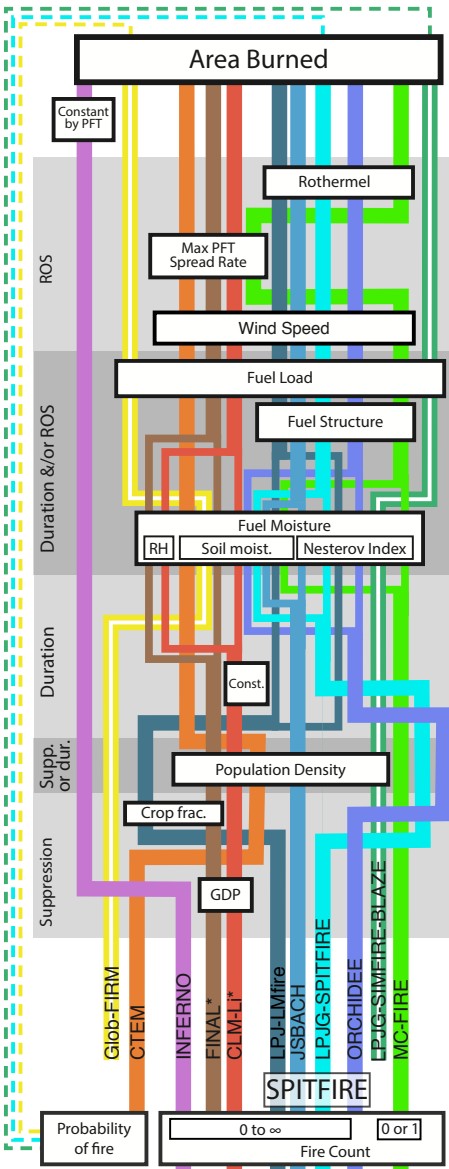

**Figure 4.** Modeled processes leading from fire starts (bottom; Fig. 3) to the calculation of burned area (top). The main processes include suppression, duration, and rate of spread ($ROS$). (Some variables can contribute to more than one of these processes; dark gray overlap areas.) "Suppression" refers to the reduction of burned area per fire. Some models apply this after the calculation of other terms (as in CLM-Li*, LM3-FINAL, LPJ-LMfire, and LPJ-GUESS-SIMFIRE-BLAZE) or it can affect fire duration (as in CTEM and JSBACH-SPITFIRE). Suppression can be a function of "GDP," crop fraction ("crop frac."), or "population density." "Fuel structure" refers to the distribution of fuel among different size classes. The "Rothermel" equations (Rothermel, 1972) are used by some models to determine rate of spread based on fire intensity and other factors. The LPJ-GUESS models convert burned area to a probability of fire (dotted lines), burning individual patches stochastically. LPJ-GUESS-GlobFIRM and LPJ-GUESS-SIMFIRE-BLAZE are denoted with white stripes to indicate that they are using purely empirical formulas to calculate gridcell-level burned area instead of simulating fire spread.

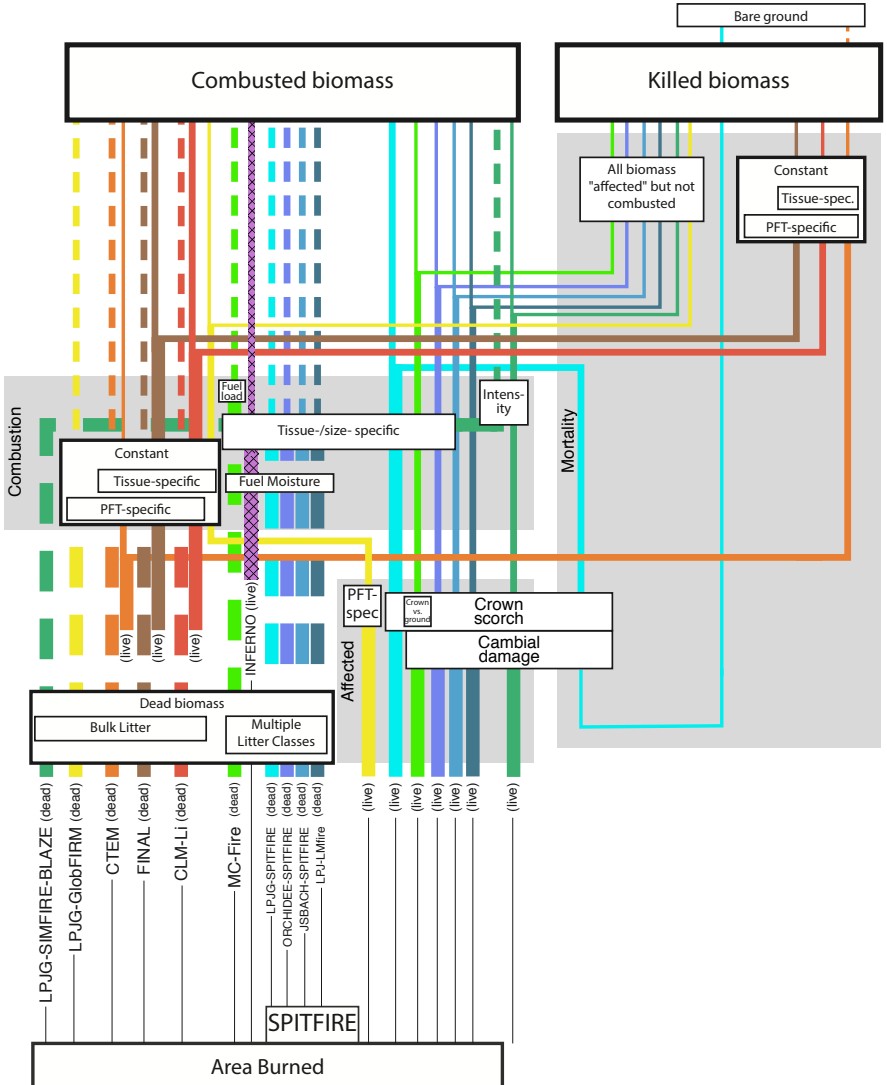

**Figure 5.** Modeled processes leading from burned area (bottom; Fig. 4) to fire combustion and mortality (top). We distinguish between combusted and killed biomass based on whether it is transferred to the atmosphere or to litter/soil pools, respectively. For live biomass, the order in which combustion and fire mortality are simulated differs among the models (Section 3); this is illustrated by the location at which lines diverge and where they are reduced in size. In some models, the amount of biomass "affected" by fire depends on simulated "crown scorch" and "cambial damage." The fraction of biomass combusted is either a constant by vegetation type ("combustion factors") or a "tissue-/size- specific" function dependent on "fuel moisture," "fuel load," and/or fire "intensity." The fraction of biomass killed is sometimes simply all affected biomass that was not combusted. In other models, constant "mortality factors" for each vegetation type give the fraction of vegetation killed in burns. LPJ-GUESS-SPITFIRE and CTEM can both then simulate the creation of "bare ground" as a result of fire death, although this will be turned off for CTEM in this phase of FireMIP (dashed line). JULES-INFERNO (cross-hatched line) does not calculate fire mortality and only calculates fire emissions diagnostically (i.e., material is not actually transferred from vegetation to the atmosphere).

**Table 1.** Experiments run in this first phase of FireMIP. All experiments used repeated (rptd.) 1901–1920 climate forcings from the beginning of the simulation through 1900. "Year 1" refers to the first transient (non-spinup) year of the simulation, which is 1700 for all models except for CLM-Li (1850) and CTEM (1861).

| Abbrv. | Name | Fire | Climate | $CO_2$ | Lightning | Pop. dens. | Land use |
|--------|------|------|---------|---------|-----------|------------|----------|
| SF1 | Transient run | On | Transient | Transient | Transient | Transient | Transient |
| SF2_WWF | World without fire | Off | Transient | Transient | Transient | Transient | Transient |
| SF2_CLI | Preindustrial climate | On | Rptd. 1901–1920 | Transient | Transient | Transient | Transient |
| SF2_CO2 | Preindustrial $CO_2$ | On | Transient | 277.33 ppm | Transient | Transient | Transient |
| SF2_FLI | Fixed lightning | On | Transient | Transient | Rptd. 1901–1920 | Transient | Transient |
| SF2_FPO | Fixed population density | On | Transient | Transient | Transient | Fixed: Year 1 | Transient |
| SF2_FLA | Fixed land use | On | Transient | Transient | Transient | Transient | Fixed: Year 1 |

**Table 2.** List of models participating in FireMIP, including contact person's email and key references. Also included is information relating to the configuration to be used in this phase of FireMIP. Note that "Resolution" refers to spatial and temporal resolution of the fire model only; the associated land/vegetation may update more frequently.

| Fire model | Land/vegetation model | Dynamic vegetation | | | N cycle? | # PFTs | # soil layers | # litter classes | Resolution | Contact |
|---|---|---|---|---|---|---|---|---|---|---|
| | | Physiology | LAI, biomass | Biogeography | | | | | | |
| CLM-Li fire module (Li et al., 2012, 2013, 2014) | CLM4.5–BGC (Oleson et al., 2013) | Yes | Yes | Yes, but in FireMIP | Yes | 17 | 15 | 1 | ~1.9° lat. × 2.5° lon. (F19), half-hourly | Fang Li (lifang@mail.iap.ac.cn) |
| CTEM fire module (Arora and Boer, 2005; Melton and Arora, 2016) | CTEM (Arora and Boer, 2005; Melton and Arora, 2016) | Yes | Yes | Yes, but not in FireMIP | No | 9 | 3 | 1 | 2.8125°, daily | Joe Melton (joe.melton@canada.ca) |
| Fire Including Natural & Agricultural Lands model (LM3-FINAL; Rabin, 2016; Rabin et al., in prep.) | LM3 (Shevliakova et al., 2009; Milly et al., 2014; Sulman et al., 2014) | Yes | Yes | Yes | No | 5 | 20 | 3 | 2° lat. × 2.5° lon., half-hourly | Dan Ward (dsward@princeton.edu) |
| INteractive Fire and Emission algoRithm for Natural envirOnments (JULES-INFERNO; Mangeon et al., 2016) | JULES (Best et al., 2011; Clark et al., 2011) | Yes | Yes | Yes, but without fire feedback | No | 9 | 4 | 4 | ~1.2414° lat. ×1.875° lon., half-hourly | Stéphane Mangeon (stephane.mangeon12@imperial.ac.uk) |
| JSBACH-SPITFIRE (Lasslop et al., 2014; Hantson et al., 2015a) | JSBACH | Yes | Yes | Yes, but not in FireMIP | No | 12 | 5 | 2 | 1.875°, daily | Gitta Lasslop (gitta.lasslop@mpimet.mpg.de) |
| LPJ-LMfire (Pfeiffer et al., 2013) | LPJ (Sitch et al., 2003) | Yes | Yes | Yes | No | 9 | 2 (plus O-horizon) | 3 | 0.5°, daily | Jed Kaplan (jed.kaplan@unil.ch) |
| LPJ-GUESS-SIMFIRE-BLAZE | LPJ-GUESS (Smith et al., 2001, 2014; Lindeskog et al., 2013) | Yes | Yes | Yes | Yes | 19 | 2 | 3 | 0.5°, annual | Stijn Hantson (stijn.hantson@kit.edu), Lars Nieradzik (lars.nieradzik@cec.lu.se) |
| LPJ-GUESS-GlobFIRM | LPJ-GUESS (Smith et al., 2001; Lindeskog et al., 2013; Smith et al., 2014) | Yes | Yes | Yes | Yes | 19 | 2 | 2 | 0.5°, annual | Stijn Hantson (stijn.hantson@kit.edu) |
| LPJ-GUESS-SPITFIRE (Lehsten et al., 2009; Thonicke et al., 2010; Lehsten et al., 2015) | LPJ-GUESS (Smith et al., 2001; Sitch et al., 2003; Ahlström et al., 2012) | Yes | Yes | Yes | No | 13 | 2 | 2 | 0.5°, daily | Matthew Forrest (matthew.forrest@senckenberg.de) |
| MC-Fire | MC2 (Bachelet et al., 2015; Sheehan et al., 2015) | Yes | Yes | Yes | Yes | 39 | Depends on total soil depth | 5 | 0.5°, monthly | Dominique Bachelet (dominique@consbio.org) |
| ORCHIDEE-SPITFIRE (Yue et al., 2014, 2015) | ORCHIDEE | Yes | Yes | Yes, but not in FireMIP | No | 13 | 2 | 2 | 0.5°, daily | Chao Yue (chao.yue@lsce.ipsl.fr) |

**Table 3.** Standard output variables. See Table A1 for additional, optional output variables. *: If calculated by model. "Crop harvesting to atmosphere" and "grazing to atmosphere" refer to carbon that is removed from the land system, but which may be emitted over an extended time period to represent the residence time of different pools.

| Category | Name | Units | Dimensions | Time period |
|---|---|---|---|---|
| Fire | Fire emissions: Total C | $\mathrm{kgC\,m^{-2}\,s^{-1}}$ | lon. lat. PFT month | 1700–2013 |
| | Fire emissions: $CO_2-C$ | $\mathrm{kgC\,m^{-2}\,s^{-1}}$ | lon. lat. month | 1700–2013 |
| | Fire emissions: $CO-C$ | $\mathrm{kgC\,m^{-2}\,s^{-1}}$ | lon. lat. month | 1950–2013 |
| | Burned fraction of gridcell | — | lon. lat. PFT month | 1700–2013 |
| | Fireline intensity* | $\mathrm{kW\,m^{-1}}$ | lon. lat. month | 1950–2013 |
| | Fuel loading | $\mathrm{kgC\,m^{-2}}$ | lon. lat. month | 1700–2013 |
| | Fuel combustion completeness | — | lon. lat. month | 1950–2013 |
| | Fuel moisture* | — | lon. lat. month | 1950–2013 |
| | Number of fires* | $\mathrm{count\,m^{-2}\,yr^{-1}}$ | lon. lat. month | 1950–2013 |
| | Fire-caused frac. tree mortality | — | lon. lat. month | 1950–2013 |
| | Fire size: Mean* | $\mathrm{m^{-2}}$ | lon. lat. month | 1950–2013 |
| | Fire size: 95th percentile | $\mathrm{m^{-2}}$ | lon. lat. month | 1950–2013 |
| Physical properties | Total soil moisture content | $\mathrm{kg\,m^{-2}}$ | lon. lat. month | 1950–2013 |
| | Total runoff | $\mathrm{kg\,m^{-2}\,s^{-1}}$ | lon. lat. month | 1950–2013 |
| | Total evapotranspiration | $\mathrm{kg\,m^{-2}\,s^{-1}}$ | lon. lat. month | 1950–2013 |
| Carbon fluxes | Gross Primary Production (grid cell) | $\mathrm{kgC\,m^{-2}\,s^{-1}}$ | lon. lat. month | 1950–2013 |
| | Gross Primary Production (by PFT) | $\mathrm{kgC\,m^{-2}\,s^{-1}}$ | lon. lat. PFT month | 1950–2013 |
| | Autotrophic respiration | $\mathrm{kgC\,m^{-2}\,s^{-1}}$ | lon. lat. month | 1950–2013 |
| | Net Primary Production (grid cell) | $\mathrm{kgC\,m^{-2}\,s^{-1}}$ | lon. lat. month | 1950–2013 |
| | Net Primary Production (by PFT) | $\mathrm{kgC\,m^{-2}\,s^{-1}}$ | lon. lat. PFT month | 1950–2013 |
| | Heterotrophic respiration | $\mathrm{kgC\,m^{-2}\,s^{-1}}$ | lon. lat. month | 1950–2013 |
| | Net Biospheric Production (grid cell) | $\mathrm{kgC\,m^{-2}\,s^{-1}}$ | lon. lat. month | 1950–2013 |
| | Net Biospheric Production (by PFT) | $\mathrm{kgC\,m^{-2}\,s^{-1}}$ | lon. lat. PFT month | 1950–2013 |
| | Land-use change C flux: To atmosphere (as $CO_2$) | $\mathrm{kgC\,m^{-2}\,s^{-1}}$ | lon. lat. month | 1950–2013 |
| | Land-use change C flux: To products | $\mathrm{kgC\,m^{-2}}$ | lon. lat. month | 1950–2013 |
| Carbon pools | Carbon in vegetation | $\mathrm{kgC\,m^{-2}}$ | lon. lat. month | 1700–2013 |
| | Carbon in aboveground litter | $\mathrm{kgC\,m^{-2}}$ | lon. lat. month | 1700–2013 |
| | Carbon in soil (incl. belowground litter) | $\mathrm{kgC\,m^{-2}}$ | lon. lat. month | 1700–2013 |
| | Carbon in vegetation, by PFT | $\mathrm{kgC\,m^{-2}}$ | lon. lat. PFT month | 1700–2013 |
| Vegetation structure | Fractional land cover of PFT | — | lon. lat. PFT year | 1700–2013 |
| | Leaf Area Index | $\mathrm{m^2\,m^{-2}}$ | lon. lat. PFT year | 1950–2013 |
| | Tree height | m | lon. lat. PFT year | 1950–2013 |

**Table 4.** Summary description of the observational datasets to be used for model evaluation. "Frequency" refers to the temporal resolution at which the analyses will be performed, which may be coarser than the native resolution of the data.

| Type | Variable | Source | Time period | Frequency | References |
|------|----------|--------|-------------|-----------|------------|
| Vegetation properties | GPP | Site-based | 1950–2006 | Snapshots | Luyssaert et al. (2007) |
| | | Site-based (FLUXNET) | Various | Monthly | Prentice Lab (2017); Davis et al. |
| | NPP | Site-based | Various | Snapshots | Olson et al. (2001); Luyssaert et al. (2007); Michaletz et al. (2014) |
| | Frac. tree, herbaceous, bare ground | ILSLCP II vegetation continuous fields | 1992–1993 | Snapshot | Hansen et al. (2000) |
| | Canopy height | ICESat GLAS | 2005 | Snapshot | Simard et al. (2011) |
| | Forest biomass | Composite of previous work adjusted with *in situ* measurements | 2000s | Snapshot | Avitabile et al. (2016) |
| Fire | # fires $yr^{-1}$, burned area per fire | MCD45 | 2003–2014 | Monthly | Archibald et al. (2013); Hantson et al. (2015b) |
| | Burned area | GFED4s | 1994–2014 | Monthly | Randerson et al. (2012); Giglio et al. (2013) |
| | | MCD45 | 2002–2014 | Monthly | Roy et al. (2008) |
| | | Fire_cci | 2005–2011 | Monthly | Alonso-Canas and Chuvieco (2015) |
| | Fuel load, combustion completeness | Site-based | Various | Snapshots | van Leeuwen et al. (2014) |
| Emissions | $CO_2$ | Site-based | 1998–2005 | Monthly | CDIAC: cdiac.ornl.gov |
| | Total C | GFAS | 2003–2015 | Monthly | Kaiser et al. (2012) |
| | $NO_2$ | OMI | 2005–2015 | Monthly | Krotkov (2013) |
| Hydrology | Runoff | Site-based | 1950–2005 | Ann. means | Dai et al. (2009) |

**Table A1.** Second-priority output variables. See Table 3 for primary model outputs. *: If calculated by model.

| Category | Name | Units | Dimensions | Time period |
|---|---|---|---|---|
| C fluxes | Crop harvesting to atmosphere | $\mathrm{kgC\,m^{-2}\,s^{-1}}$ | lon. lat. year | 1950–2013 |
| | Grazing to atmosphere* | $\mathrm{kgC\,m^{-2}\,s^{-1}}$ | lon. lat. year | 1950–2013 |
| | Litter to soil | $\mathrm{kgC\,m^{-2}\,s^{-1}}$ | lon. lat. year | 1950–2013 |
| | Vegetation to litter | $\mathrm{kgC\,m^{-2}\,s^{-1}}$ | lon. lat. year | 1950–2013 |
| | Vegetation to soil | $\mathrm{kgC\,m^{-2}\,s^{-1}}$ | lon. lat. year | 1950–2013 |
| Fire | Ignitions* | $\mathrm{m^{-2}\,yr^{-1}}$ | lon. lat. month | 1950–2013 |
| Physical properties | Broadband albedo (by PFT) | — | lon. lat. PFT month | 1950–2013 |
| | Evaporation: Canopy | $\mathrm{kg\,m^{-2}\,s^{-1}}$ | lon. lat. year | 1950–2013 |
| | Evaporation: Soil | $\mathrm{kg\,m^{-2}\,s^{-1}}$ | lon. lat. year | 1950–2013 |
| | Evaporation: Soil (by PFT) | $\mathrm{W\,m^{-2}}$ | lon. lat. PFT month | 1950–2013 |
| | Evapotranspiration (by PFT) | $\mathrm{W\,m^{-2}}$ | lon. lat. PFT month | 1950–2013 |
| | Near-surface air temperature | K | lon. lat. year | 1950–2013 |
| | Net radiation (by PFT) | $\mathrm{W\,m^{-2}}$ | lon. lat. PFT month | 1950–2013 |
| | Irrigation (by PFT) | $\mathrm{kg\,m^{-2}\,s^{-1}}$ | lon. lat. PFT year | 1950–2013 |
| | Precipitation | $\mathrm{kg\,m^{-2}\,s^{-1}}$ | lon. lat. year | 1950–2013 |
| | Sensible heat flux (by PFT) | $\mathrm{W\,m^{-2}}$ | lon. lat. PFT month | 1950–2013 |
| | Skin temperature (by PFT) | K | lon. lat. PFT year | 1950–2013 |
| | Snow depth or equivalent (by PFT) | $\mathrm{m\,m^{-2}}$ | lon. lat. PFT month | 1950–2013 |
| | Soil moisture (by PFT) | $\mathrm{kg\,m^{-2}}$ | lon. lat. PFT year | 1950–2013 |
| | Soil temperature | K | lon. lat. layer year | 1950–2013 |
| | Surface downwelling shortwave radiation | $\mathrm{W\,m^{-2}}$ | lon. lat. year | 1950–2013 |
| | Transpiration | $\mathrm{kg\,m^{-2}\,s^{-1}}$ | lon. lat. year | 1950–2013 |
| | Transpiration (by PFT) | $\mathrm{W\,m^{-2}}$ | lon. lat. PFT month | 1950–2013 |
| Vegetation structure | Leaf area index | $\mathrm{m^2\,m^{-2}}$ | lon. lat. year | 1950–2013 |

*Author contributions.* All authors contributed to the development of the protocol, with A. Arneth and S. Hantson leading and contributing text for Section 2. S. Rabin compiled and edited text from other authors, wrote the Introduction and Conclusion, and constructed the tables (with help from co-authors listed below). S. Sitch contributed text for Section 2.2. S. Harrison contributed to the Evaluation section. J. Melton and S. Rabin constructed Figures 3–5. G. Lasslop performed analyses for and contributed Figure 1. J. Kaplan constructed lightning dataset. V. Arora, D. Bachelet, M. Forrest, T. Hickler, J. Kaplan, S. Kloster, W. Knorr, G. Lasslop, F. Li, J. Melton, S. Mangeon, L. Nieradzik, S. Rabin, A. Spessa, D. Ward, and C. Yue contributed text and information for model descriptions, tables, and flowchart figures, and contributed to model development. G. Folberth, T. Sheehan, and A. Voulgarakis contributed to model development. D. Kelley helped design the analytical protocol.

*Acknowledgements.* S. Rabin was supported by a National Science Foundation Graduate Research Fellowship and by the Carbon Mitigation Initiative, and along with S. Hantson and A. Arneth would like to acknowledge support by the EU FP7 projects BACCHUS (grant agreement no. 603445) and LUC4C (grant agreement no. 603542). This work was supported, in part, by the German Federal Ministry of Education and Research (BMBF), through the Helmholtz Association and its research programme ATMO, and the HGF Impulse and Networking fund. F. Li was funded by the National Natural Science Foundation of China under Grant No. 41475099 and the CAS Youth Innovation Promotion Association Fellowship. The UK Met Office contribution was funded by BEIS under the Hadley Centre Climate Programme contract (GA01101). G. A. Folberth also wishes to acknowledge funding received from the European Union's Horizon 2020 research and innovation programme under grant agreement No 641816 (CRESCENDO). J. O. Kaplan was supported by the European Research Council (COEVOLVE, 313797).

The article processing charges for this open-access publication were covered by a Research Center of the Helmholtz Association. We acknowledge support from the Deutsche Forschungsgemeinschaft and the Open Access Publishing Fund of the Karlsruhe Institute of Technology.

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
