# Peer review of "The Fire Modeling Intercomparison Project (FireMIP), phase 1: Experimental and analytical protocols with detailed model descriptions"

_Geoscientific Model Development, 2016_

## Short Comment (SC1) · 24 Oct 2016

Dear authors,

In my role as Executive editor of GMD, I would like to bring to your attention our Editorial version 1.1:

http://www.geosci-model-dev.net/8/3487/2015/gmd-8-3487-2015.html

This highlights some requirements of papers published in GMD, which is also available on the GMD website in the 'Manuscript Types' section:

http://www.geoscientific-model-development.net/submission/manuscript_types.html

In particular, please note that for your paper, the following requirements have not been met in the Discussions paper:

- "Inclusion of Code and/or data availability sections is mandatory for all papers and should be located at the end of the article, after the conclusions, and before any appendices or acknowledgments. For more details refer to the code and data policy" (Editorial v1.1, Appendix A1)

Please first move the data availability section to the correct place in your revised submission to GMD. Note that it is not a numbered section. Additionally, please use this section to inform the reader where the data will be placed. Just telling that it will be made available is not enough.

Yours,

Astrid Kerkweg

---

## Short Comment (SC2) · 13 Nov 2016

General comments The manuscript provides a well written overview of available global fire models, which is a really valuable contribution to the literature, highly relevant to the journal GMD. The level of detail about each of the models is excellent, with specific equations included to describe how each model is parameterised –it is great to see this sort of information in one place. The relevant references are included, so if more detail were required, it would be easy to find. However, since one of the main aims of the manuscript is to describe Phase 1 of an Intercomparison Project, there needs to be much more detail in the sections describing Model evaluation, Benchmarking, empirical and observational comparisons (more detail below). Is there any reason why

the manuscript is split into Phase 1 and doesn't present the results of the evaluations? Is it that the evaluations have not yet been completed, or that it would just be too much for one paper?

Specific comments Page 2, line 17 – Define fire regime somewhere in the Introduction It would be good to be told somewhere in the text that the models include PFTs relevant to both the Northern and Southern hemisphere conditions. As it is, the reader has to go to Table S15 in the Supplementary Material before finding out, for example, which models include evergreen vegetation. A table listing the FireMIP prescriptions should be included. For example, what is the prescribed vegetation height? What other prescriptions are there? Justify why they are imposed. Page 5, line 17 –Why were the 1901–1920 climate and 20 lightning inputs recycled for the first 200 years of the simulation? Page 6, line 3 – Write PFT in full the first time it's used Page 10, line 1 – Define GFED regions Page 10 – Model evaluation, Benchmarking, empirical and observational comparisons. These sections need a lot more detail, considering that the aim of the paper is to describe the experimental and analytical protocols. The benchmarking paragraph seems overly simplistic, and could be improved by a more thoughtful consideration of the difficulties in validating model output or comparing models. What sort of observations will be used to assess model performance? There is very little detail given in the table. What are the "appropriate tools" that will be used? The identification of causal relationships is notoriously difficult in interactions between climate, weather, vegetation and fire, so it's important to say how you're going to assess the models. Page 11, line 20 – when and where will the data be made available?

Technical corrections Page 3, line 18 – the apostrophes around "right output" and "wrong reasons" are unnecessary Page 4, line 21 – delete "of" before "an" Page 9, line 3 – delete "been" Page 12, line 22 – expand eg. to words, "including, for example,..." Page 12, line 33 –delete "that" Page 14, figure legend, line 3 – delete period before bracket Page 15, figure legend, line 2 – delete period before bracket

---

## Referee Comment (RC1) · Anonymous Referee #1 · 15 Dec 2016

The paper 'the fire modeling intercomparison project (fire MIP), phase 1: experimental and analytical protocols" by Rabin et al. aimes at synthesizing the various fire models embedded in DGVMs, describe their processes and equations, and how we can compare them. The complete description of all the equations in one single synthesis is definitely a keystone information for the community and is very well presented. The benchmarking datasets available and the efforts for proposing a common simulation protocol to capture the differences are clearly explained and is promising for further understanding on this topic. the paper is well written to try and clearly synthesize all the models. In turn, I just have minor concerns on the few 6 points below: #1# Page 3 lines 15 to 20 in introduction: I think some keystone references on the long term data

or simulations performed over the last century are missing here, beside the Kasischke 2002 and Stock 2003 references. I think it would be worth mentionning some model benchmarking (Yue et al. 2015, Kloster et al. 10), some fire history records and emissions (Mouillot et al. 2005, 2006, Schultz et al. 2008, Mieville et al. 2010), and the recent synthesis on global charcoal database to be used potentially for recent trends (Marlon et al. 2016). #2#for the LPJ GUESS BLAZE description, i did not really understand 'this annual burned area is distributed to each month of the next year based on observed fire seasonality'... i am confused with the term 'next year', and 'observed'. observations are based on remote sensing data? if yes, which one? and what s the impact on the benchmarking of seasonality if it's fitted on a given remote sensing data. In this sense, I think we would need a full table where, for each model, the reader would like to know for which variable and for which time step the model output can be benchmarked. #3# In table 4 describing the variables used for model benchmarking, I am wondering if fire size distribution or fire number could be an option or not? for exemple Hantson et al 2016, Yue et al. 2015 started to use this variable, and Oom et al. 2016 (recently published in Remote Sensing, maybe after the final submission of this manucript) proposed a global database on these fire numbers. #4# in the datasets description, we get a little confused along the document on the different time frames... maybe an additional supplementary material would help in understanding what are the actual data time frames, and the time frames for which the authors have repeated some variables. #5# finally, in table S3, what is the difference between 'none' and 'n/a'? in this supplementary material, n/a, n/c and none should be more clearly defined. #6# in table A1: I could see the "grazing to atmosphere" variable. this is not well described in the models, and which model actually use this. any additional information to provide on this topic?

---

## Author Comment (AC1) · 30 Jan 2017

We thank the editors and reviewers for their help in improving this manuscript. Attached, find a ZIP file with new versions of the main text and supplementary PDF. Included as well is a PDF containing our responses to reviewers, a list of miscellaneous other changes, and the output of the latexdiff command between the original and new versions of the main text.

Please also note the supplement to this comment: http://www.geosci-model-dev-discuss.net/gmd-2016-237/gmd-2016-237-AC1-supplement.zip

---

## Author Response (AR1)

**Response to Dr. Harris**

We thank Dr. Harris for her kind comments and constructive criticism, our responses to which are below.

**General comments**

*The manuscript provides a well written overview of available global fire models, which is a really valuable contribution to the literature, highly relevant to the journal GMD. The level of detail about each of the models is excellent, with specific equations included to describe how each model is parameterized – it is great to see this sort of information in one place. The relevant references are included, so if more detail were required, it would be easy to find. However, since one of the main aims of the manuscript is to describe Phase 1 of an Intercomparison Project, there needs to be much more detail in the sections describing Model evaluation, Benchmarking, empirical and observational comparisons (more detail below).*

*Is there any reason why the manuscript is split into Phase 1 and doesn't present the results of the evaluations? Is it that the evaluations have not yet been completed, or that it would just be too much for one paper?*

We wanted to publish the Phase 1 protocol before the analysis of the results for several reasons. Primarily, we would like for as many modeling groups as possible to be a part of the FireMIP effort, and publishing now rather than waiting for all analyses to be complete allows more groups to come on board early. Another consideration is that a large number of analyses with many different aims are being planned for the Phase 1 runs, which will span several different papers. We believe it makes more sense for the protocol to stand on its own, rather than to be arbitrarily tied to any one of the analysis papers.

**Specific comments**

*Page 2, line 17 – Define fire regime somewhere in the Introduction*
This now occurs in the first paragraph: page 2, lines 25–26. New text here in bold:
> Mitigating the most harmful consequences of changing fire regimes – **the typical pattern of fire occurrence as characterized by frequency, seasonality, size, intensity, and ecosystem effects, among other factors (Pyne et al., 1996)** – could require new strategies for managing ecosystems (Moritz et al., 2014).

*It would be good to be told somewhere in the text that the models include PFTs relevant to both the Northern and Southern hemisphere conditions. As it is, the reader has to go to Table S15 in the Supplementary Material before finding out, for example, which models include evergreen vegetation.*
The first few sentences of the last paragraph in Section 2.3 have been edited to introduce the idea of plant functional types. Original:
> The biogeography of natural vegetation, represented by plant functional types, was either prescribed by modeling groups or simulated dynamically.

New (now p. 6, lines 5–7):

> The biogeography of natural vegetation, represented by plant functional types (major global vegetation classes; PFTs), was either prescribed by modeling groups or simulated dynamically.

*A table listing the FireMIP prescriptions should be included. For example, what is the prescribed vegetation height? What other prescriptions are there? Justify why they are imposed.*

It is somewhat unclear what is being referred to here. As much as possible, PFT-specific prescriptions for each model are described in the supplementary tables. For example, PFT-specific woody vegetation height for ORCHIDEE-SPITFIRE is available in Table S24.

For clarity, the following sentence has been added to what is now the third paragraph in Section 3 (p. 7, lines 9–11):

> We have also included PFT-specific parameters and equations in Tables S16–S26; these were prescribed by the modeling groups during the development of their respective fire models either due to limitations of their vegetation models or intentionally based on development plans and priorities.

*Page 5, line 17 – Why were the 1901–1920 climate and 20 lightning inputs recycled for the first 200 years of the simulation?*

CRU-NCEP and lightning forcings are not available for 1701–1900. We chose to recycle the 1901–1920 forcings over that period because 20 years is long enough to capture decadal-scale natural climate variability, while 1920 is early enough that minimal human influence on climate can be expected (and thus it should be representative of the climate back to 1701).

The first two paragraphs of Section 2.3 (p. 5 L. 26 – p. 6 L. 7) have been edited to explain this and to generally improve clarity. A new figure (Fig. 2, p. 17) has also been added to clarify the time periods involved with the input data for model runs.

*Page 6, line 3 – Write PFT in full the first time it's used*

This now occurs near the end of Section 2.3 (p. 6 LL. 5–7).

*Page 10, line 1 – Define GFED regions*

A definition of the term and a citation have been added (p. 11, L. 27).

*Page 10 – Model evaluation, Benchmarking, empirical and observational comparisons. These sections need a lot more detail, considering that the aim of the paper is to describe the experimental and analytical protocols. The benchmarking paragraph seems overly simplistic, and could be improved by a more thoughtful consideration of the difficulties in validating model output or comparing models.*

Section 4.1 has been revised and expanded (p. 12) to better explain the reason this benchmarking protocol has been chosen, as well as to discuss its limitations with regards to actually diagnosing the causes of performance differences between models. Text has been added to Section 4.2 (p. 13, L. 1–13) to clarify that the FireMIP protocol calls for the use of generalized linear models for assessing model sensitivities and abilities to recreate emergent patterns. Section 4.3 has been expanded (p. 13, L. 29–33) to discuss limitations on benchmarking imposed by data quality and availability.

*What sort of observations will be used to assess model performance? There is very little detail given in the table.*

Table 4 describes the observational data that will be used in the formal benchmarking protocol described in our Section 4.1 and outlined in detail by Kelley et al. (2013). Details in Table 4 include the source of the data (satellite product name or "site-based" for field observations), the time period covered, the temporal resolution of the data (or "snapshots" for one-off measurements), and key references for further investigation. We believe this gives the reader a good sense of the breadth and types of data that we will be using for benchmarking.

As far as the benchmarking protocol itself, we decided to leave out the mathematical details (a) in favor of succinctness and (b) in the interest of not simply repeating what is already thoroughly described by Kelley et al. (2013). The interested reader can peruse that article for details.

*What are the "appropriate tools" that will be used? The identification of causal relationships is notoriously difficult in interactions between climate, weather, vegetation and fire, so it's important to say how you're going to assess the models.*

A paragraph has been added to Section 4.1 (p. 12, L. 25–31), explaining that the intention of FireMIP phase 1 is not necessarily to identify causal relationships per se, but rather to serve as a jumping-off point for more targeted experiments. It might be possible to draw some conclusions based on the results of the SF2 experiments, but undoubtedly there will be limitations.

**Page 11, line 20 – when and where will the data be made available?**
The Data Availability section (p. 15, L. 15–18), which has been moved to its proper location after the Discussion & Conclusions, has been changed to read:

> Once all runs are completed, model outputs will be made available to the public at https://bwfilestorage.lsdf.kit.edu/public/projects/imk-ifu/FireMIP. The FireMIP website (http://www.imk-ifu.kit.edu/firemip.php) will also be kept up-to-date with any changes to data access procedure, in addition to project updates and summary information.

**Technical corrections**

The following changes have been applied (old locations):
- *Page 3, line 18 – the apostrophes around "right output" and "wrong reasons" are unnecessary*
- *Page 4, line 21 – delete "of" before "an"*
- *Page 9, line 3 – delete "been"*
- *Page 12, line 22 – expand eg. to words, "including, for example, ..."*
- *Page 12, line 33 – delete "that"*

Two suggested changes have not been applied (old locations):
- *Page 14, figure legend, line 3 – delete period before bracket*
- *Page 15, figure legend, line 2 – delete period before bracket*

Because the bracketed clauses are complete sentences, we have written them independently of their preceding sentences, which then need to be ended with periods.

**Response to Anonymous Reviewer**

We thank the anonymous reviewer for their kind comments and helpful suggestions.

*Page 3 lines 15 to 20 in introduction: I think some keystone references on the long term data or simulations performed over the last century are missing here, beside the Kasischke 2002 and Stock 2003 references. I think it would be worth mentionning some model benchmarking (Yue et al. 2015, Kloster et al. 10), some fire history records and emissions (Mouillot et al. 2005, 2006, Schultz et al. 2008, Mieville et al. 2010), and the recent synthesis on global charcoal database to be used potentially for recent trends (Marlon et al. 2016).*

        The suggested citations have been added. Original text:

            There are regional compilations of data from other sources, of varying quality, that extend back to the mid-20th century (Kasischke et al., 2002; Stocks et al., 2003). Both types of observational records will be used to evaluate model performance, but the first half of the twentieth century is quite data-poor.

        New (p. 3 LL. 20–26):

            Charcoal records do not yet have global coverage, and there are uncertainties even in trend for the twentieth century (Marlon et al., 2016). Literature reviews, sometimes in combination with regional burned area statistics extending back to the 1960s (Kasischke et al., 2002; Stocks et al., 2003) and/or simulation models, have been used to produce estimates of burned area and associated emissions going back to the beginning of the twentieth century (Mouillot and Field, 2005; Mouillot et al., 2006; Schultz et al., 2008; Mieville et al., 2010). Both remote sensing data and historical reconstructions can be used to evaluate model performance, but the pre-1990s period – especially before the 1960s – is quite data-poor.

*For the LPJ GUESS BLAZE description, i did not really understand 'this annual burned area is distributed to each month of the next year based on observed fire seasonality'... i am confused with the term 'next year', and 'observed'. observations are based on remote sensing data? if yes, which one?*

        The LPJ-GUESS-SIMFIRE-BLAZE model description has been corrected and revised for clarity. Original text:

            This annual burned area is distributed to each month of the next year based on observed fire seasonality.

        New text (p. 10, LL. 12–13):

            This annual burned area is distributed to each month of the year based on mean observed seasonality of burned area from GFED3 (Giglio et al., 2010).

*and what s the impact on the benchmarking of seasonality if it's fitted on a given remote sensing data. In this sense, I think we would need a full table where, for each model, the reader would like to know for which variable and for which time step the model output can be benchmarked.*

        If a model is based on remote sensing data, as LPJ-GUESS-SIMFIRE-BLAZE is with regard to seasonal timing, then it would show perfect correspondence with the observations in the benchmarking. However, to avoid confusion with models that generate their own seasonality, LPJ-GUESS-SIMFIRE-BLAZE will be excluded from seasonality benchmarking.

*In table 4 describing the variables used for model benchmarking, I am wondering if fire size distribution or fire number could be an option or not? for exemple Hantson et al 2016, Yue et al. 2015 started to use this variable, and Oom et al. 2016 (recently published in Remote Sensing,*

*maybe after the final submission of this manucript) proposed a global database on these fire numbers.*

Burned area per fire and number of fires have been added to Table 4 as benchmark variables, with citations to Archibald et al. (2013) and Hantson et al (2015b) as the datasets being used.

*In the datasets description, we get a little confused along the document on the different time frames... maybe an additional supplementary material would help in understanding what are the actual data time frames, and the time frames for which the authors have repeated some variables.*

A figure has been added to clarify the timelines involved with the different input datasets for the spinup and historical runs (now Figure 2).

*Finally, in table S3, what is the difference between 'none' and 'n/a'? in this supplementary material, n/a, n/c and none should be more clearly defined.*

Clarification on this point has been added throughout the Supplementary Tables.

*In table A1: I could see the "grazing to atmosphere" variable. this is not well described in the models, and which model actually use this. any additional information to provide on this topic?*

Every model that simulates grazing is expected to potentially produce this output variable (although, since it is a second-priority variable, it is an optional output). The following explanation has been added to the caption of Table A1:

"Crop harvesting to atmosphere" and "grazing to atmosphere" refer to carbon that is removed from the land system, but which may be emitted over an extended time period to represent the residence time of different pools.

**Additional changes and corrections**

The title has been changed to (new part in bold): "The Fire Modeling Intercomparison Project (FireMIP), phase 1: Experimental and analytical protocols **with detailed model descriptions**."

The LM3-FINAL and LPJ-LMfire models have been added, with concomitant additions to text, Figures 3–5 (formerly 2–4), and tables. Daniel Ward (Princeton University, LM3-FINAL) has been added to the author list and contributions section; Jed Kaplan (University of Lausanne, LPJ-LMfire) has been moved up in the author list.

Douglas Kelley has been added to the author list.

All references to LPJ-GUESS-BLAZE have been changed to LPJ-GUESS-SIMFIRE-BLAZE.

INFERNO is now referred to as JULES-INFERNO for consistency with other model names.

*Main text*

The last sentence of the abstract has been replaced. Original:
> Here we introduce the fire models used in the first phase of FireMIP, the simulation protocols applied, and the benchmarking system used to evaluate the models.

New (p. 2, L. 11):
> In this paper, we introduce the fire models used in the first phase of FireMIP, the simulation protocols applied, and the benchmarking system used to evaluate the models. We have also created supplementary tables that describe, in thorough mathematical detail, the structure of each model.

A citation of Carvalho et al. (2010) in the Introduction (now p. 2, L. 20) has been corrected to refer to a different paper, Carvalho et al. (2011).

The sentence now at p. 2 LL. 28–30 has been changed from:
> However, that analysis relied on one statistical model of fire that was forced with a number of different climate projections; the effects of increased atmospheric carbon dioxide, changes in vegetation productivity and structure, and fire-vegetation-climate feedbacks were not considered.

To (changed text in bold):
> However, that analysis **largely relied on statistical models of fire danger and burned area, forced with a number of different climate projections**; the effects of increased atmospheric carbon dioxide, changes in vegetation productivity and structure, and fire-vegetation-climate feedbacks were not considered.

The beginning of the sentence now at p. 3 L. 17 has been changed from, "Direct observations on fire occurrence" to, "Direct observations **of** fire occurrence".

The beginning of the sentence now at p. 3 LL. 26–27 has been changed from, "This first phase of FireMIP will thus serve to produce a sort of ensemble estimate of global fire activity during that time" to, "This first phase of FireMIP will thus serve to produce **an** estimate of global fire activity during that time".

A sentence has been added to the end of the first paragraph in Section 2.3 (now p. 5, LL. 29–30): "Note that for various reasons some modeling groups may not be able to use 1700 CE as the beginning of their run, with CLM-Li preferring 1850 and CTEM preferring 1861."

A reference to Table 2 has been added at the end of Section 2.3 (p. 6, L. 7).

Three uses of the past tense in Section 2.4 have been changed to present tense (p. 6, LL. 9–13).

The third sentence of Section 2.4 has been changed from:
> All gridded outputs are provided in NetCDF format at at least 0.5º resolution or on the native grid of the model if run at a coarser resolution.

To (p. 6, LL. 10–11):
> All outputs are to be provided in NetCDF format at the native spatial resolution of the model, and at either monthly or annual temporal resolution (Tables 3, A1).

The beginning of the first sentence in Section 3 has been changed from, "Nine models have run" to, "**Eleven** models **are running**" (now p. 6, L. 15).

The second sentence of Section 3 has been changed from
> All simulate fire in "natural" ecosystems, with some also simulating cropland, pasture, deforestation, and peat fire (Table S3).

To (p. 6, LL. 15–17):
> All simulate fire in ``natural" ecosystems, which are composed of a variety of PFTs representing major vegetation classes around the world. Some models also simulate cropland, pasture, deforestation, and peat fire (Table S3).

The end of the last sentence of the first paragraph in Section 3 has been changed from "vary fractional mortality and combustion based on estimated fire intensity," to, "vary fractional mortality and combustion based on estimated fire intensity, **PFT-specific plant architecture and fire resistance, and other factors**." (now p. 6, LL. 27–28)

A paragraph about the order in which combustion and mortality are carried out has been added to Section 3 (p. 6 L. 29 to p. 7 L. 3).

A reference to Figures 3–5 has been added to Section 3 (now p. 7, L. 11).

The last two sentences of the third paragraph in Section 3 have been changed from:
> It should be noted, however, that most of these models are still under continuous development. Readers should thus not assume that the descriptions given here are applicable to anything except the model versions used for this phase of FireMIP.

To (p. 7, LL. 14–16):
> It should be noted, however, that most of these models are under continuous development; it should not be assumed that the descriptions given here apply to anything except the model versions used for this phase of FireMIP.

The model description text and tables for CTEM has been amended to reflect changes to the code since its most recent documentation.

A description of the LM3-FINAL model has been added (now p. 8 L. 28 to p. 9 L. 11).

A description of the LPJ-LMfire model has been added (now p. 9 LL. 13–27).

A sentence in the LPJ-GUESS-GlobFIRM model description (now Section 3.7) has been changed from:
> (Note that because LPJ-GUESS-GlobFIRM estimates burned area directly, no outputs having to do with number of fires or fire size will be generated.)

to (now p. 9, LL. 32–33):
> (As LPJ-GUESS-GlobFIRM estimates burned area directly, it does not generate outputs of fire count or size.)

An additional citation has been added to the LPJ-GUESS-SPITFIRE model description, regarding PFT-specific parameterization (p. 10, L. 30).

Corrections have been made to the ORCHIDEE-SPITFIRE model description paragraph (p. 11, LL. 21–31) and tables, regarding human suppression of lightning-ignited fires.

A description of how differences in model performance can be quantified has been added to Section 4.1 (p. 12 LL. 17–20).

The beginning of a sentence in Section 4.2 has been changed from, "Model outputs are then interrogated" to, "Model outputs **can then be** interrogated" (now p. 13, L. 9).

In the Discussion and Conclusions (now Section 5), "could lead to differences in the simulated fire regimes" has been changed to "**will** lead to differences in the simulated fire regimes" (p. 14, LL. 12–13).

In that same paragraph, the beginning of a sentence has been changed from:
> The outputs from each model about leaf area and the fractional cover of different plant functional types (Table 3) for each grid cell

To (p. 14, LL. 15–16):
> Outputting information on leaf area and fractional cover of different PFTs (Table 3)

The penultimate paragraph in the Discussion and Conclusions (now Section 5) has been changed from:
> Nine modeling groups are performing the baseline FireMIP simulations, meaning that there are a number of fire models that are not included in this preliminary exercise. However, we hope that publishing this experimental and benchmarking protocol will encourage other fire modeling groups to participate in FireMIP.

To:
> **Eleven** modeling groups are performing the baseline FireMIP simulations, **but there are several other fire models in use. We** hope that publishing this experimental and benchmarking protocol will encourage other fire modeling groups to participate in FireMIP.

Figure 1 has been slightly edited for aesthetic reasons.

The flowchart figures and captions (now Figs. 3–5) have been reworked to reflect the inclusion of LM3-FINAL and LPJ-LMfire, as well as various corrections.

Figure 5 (formerly Figure 4) has been reworked to reflect the order of combustion and fire-induced mortality.

The description of JULES-INFERNO in Table 2 has been amended to indicate that simulated fires do not have any impact on biogeography.

Changes have been made to Table 4, reflecting evolution of the group of datasets to be used in the benchmarking analyses.

(Other minor changes may have escaped our notice.)

**Supplementary tables & figures**

Various changes to model equations in supplementary tables (with accompanying changes to symbols and glossary).

Two supplementary figures have been added. Figure S1 shows maps of monthly ignitions per person for JSBACH-SPITFIRE and LPJ-GUESS-SPITFIRE. Figure S2 shows a map of the GFED-region scaling parameter used in ORCHIDEE-SPITFIRE

**The Fire Modeling Intercomparison Project (FireMIP), phase 1: Experimental and analytical protocols with detailed model descriptions**

Sam S. Rabin[1,2], Joe R. Melton[3], Gitta Lasslop[4], Dominique Bachelet[5,6], Matthew Forrest[7], Stijn Hantson[2], Jed O. Kaplan[8], Fang Li[9], Stéphane Mangeon[10], Daniel S. Ward[11], Chao Yue[12], Vivek K. Arora[13], Thomas Hickler[7,14], Silvia Kloster[4], Wolfgang Knorr[15], Lars Nieradzik[16,17], Allan Spessa[18], Gerd A. Folberth[19], Tim Sheehan[6], Apostolos Voulgarakis[10], Douglas I. Kelley[20], I. Colin Prentice[21,22], Stephen Sitch[23], Sandy Harrison[24], and Almut Arneth[2]

[1]Dept. of Ecology & Evolutionary Biology, Princeton University, Princeton, NJ, USA
[2]Karlsruhe Institute of Technology, Institute of Meteorology and Climate Research / Atmospheric Environmental Research, 82467 Garmisch-Partenkirchen, Germany
[3]Climate Research Division, Environment and Climate Change Canada, Victoria, BC, V8W 2Y2, Canada
[4]Land in the Earth System, Max Planck Institute for Meteorology, Bundesstrasse 53, 20146 Hamburg, Germany
[5]Biological and Ecological Engineering, Oregon State University, Corvallis, OR 97331, USA
[6]Conservation Biology Institute, 136 SW Washington Ave., Suite 202, Corvallis, OR 97333, USA
[7]Senckenberg Biodiversity and Climate Research Institute (BiK-F), Senckenberganlage 25, 60325 Frankfurt am Main, Germany
[8]Institute of Earth Surface Dynamics, University of Lausanne, 4414 Géopolis Building, 1015 Lausanne, Switzerland
[9]International Center for Climate and Environmental Sciences, Institute of Atmospheric Physics, Chinese Academy of Sciences, Beijing, China
[10]Department of Physics, Imperial College London, London, United Kingdom
[11]Program in Atmospheric and Oceanic Sciences, Princeton University, Princeton, NJ, USA
[12]Laboratoire des Sciences du Climat et de l'Environnement, LSCE/IPSL, CEA-CNRS-UVSQ, Université Paris-Saclay, F-91198 Gif-sur-Yvette, France.
[13]Canadian Centre for Climate Modelling and Analysis, Environment and Climate Change Canada, Victoria, BC, V8W 2Y2, Canada
[14]Department of Physical Geography, Goethe-University, Altenhöferallee 1, 60438 Frankfurt am Main, Germany
[15]Department of Physical Geography and Ecosystem Science, Lund University, 22362 Lund, Sweden
[16]Centre for Environmental and Climate Research, Lund University, 22362 Lund, Sweden
[17]CSIRO Oceans and Atmosphere, PO Box 3023, Canberra, ACT 2601, Australia
[18]School of Environment, Earth and Ecosystem Sciences, Open University, Milton Keynes, UK
[19]UK Met Office Hadley Centre, Exeter, UK
[20]Centre for Ecology and Hydrology, Maclean building, Crowmarsh Gifford, Wallingford, Oxfordshire, OX10 8BB, UK
[21]School of Biological Sciences, Macquarie University, North Ryde, NSW 2109, Australia
[22]AXA Chair of Biosphere and Climate Impacts, Grand Challenges in Ecosystem and the Environment, Department of Life Sciences and Grantham Institute – Climate Change and the Environment, Imperial College London, Silwood Park Campus, Buckhurst Road, Ascot SL5 7PY, UK
[23]College of Life and Environmental Sciences, University of Exeter, Exeter EX4 4RJ, UK
[24]School of Archaeology, Geography and Environmental Sciences (SAGES), University of Reading, Reading, UK

*Correspondence to:* Sam S. Rabin (sam.rabin@kit.edu)

**Abstract.** The important role of fire in regulating vegetation community composition and contributions to emissions of greenhouse gases and aerosols make it a critical component of dynamic global vegetation models and Earth system models. Over two decades of development, a wide variety of model structures and mechanisms have been designed and incorporated into global fire models, which have been linked to different vegetation models. However, there has not yet been a systematic examination of how these different strategies contribute to model performance. Here we describe the structure of the first phase of the Fire Model Intercomparison Project (FireMIP), which for the first time seeks to systematically compare a number of models. By combining a standardized set of input data and model experiments with a rigorous comparison of model outputs to each other and to observations, we will improve the understanding of what drives vegetation fire, how it can best be simulated, and what new or improved observational data could allow better constraints on model behavior.  In this paper, we introduce the fire models used in the first phase of FireMIP, the simulation protocols applied, and the benchmarking system used to evaluate the models. We have also created supplementary tables that describe, in thorough mathematical detail, the structure of each model.

The works published in this journal are distributed under the Creative Commons Attribution 3.0 License. This license does not affect the Crown copyright work, which is re-usable under the Open Government Licence (OGL). The Creative Commons Attribution 3.0 License and the OGL are interoperable and do not conflict with, reduce, or limit each other.

© Crown Copyright 2016.

[revised manuscript text omitted]